# On the Periodic Behavior of Neural Network Training with Batch Normalization and Weight Decay

**Ekaterina Lobacheva**[1]*, **Maxim Kodryan**[1]*, **Nadezhda Chirkova**[1]
**Andrey Malinin**[1,2], **Dmitry Vetrov**[1,3]
[1]HSE University    [2]Yandex    [3]AIRI
Moscow, Russia
elobacheva@hse.ru, mkodryan@hse.ru, nchirkova@hse.ru
am969@yandex-team.ru, dvetrov@hse.ru

## Abstract

Training neural networks with batch normalization and weight decay has become a common practice in recent years. In this work, we show that their combined use may result in a surprising periodic behavior of optimization dynamics: the training process regularly exhibits destabilizations that, however, do not lead to complete divergence but cause a new period of training. We rigorously investigate the mechanism underlying the discovered periodic behavior from both empirical and theoretical points of view and analyze the conditions in which it occurs in practice. We also demonstrate that periodic behavior can be regarded as a generalization of two previously opposing perspectives on training with batch normalization and weight decay, namely the equilibrium presumption and the instability presumption.

## 1 Introduction

Normalization approaches, such as batch or layer normalization, have become vital for the successful training of modern deep neural networks [12, 2, 24, 21, 27]. Despite much recent work [3, 22, 9, 28], it is still not completely understood how normalization influences the training process. In this work, we investigate the surprising periodic behavior that may occur when a neural network is trained with a commonly used combination of some kind of normalization, in our case batch normalization (BN) [12], and weight decay regularization (WD). Examples of this behavior are provided in Figure 1.

The dynamics of neural network training with BN and WD have been examined extensively in literature due to the non-trivial competing influence of BN and WD on the norm of neural network's weights. More precisely, using BN makes (a part of) neural network's weights *scale-invariant*, i.e., multiplying them by a positive constant does not change the network's output. Although scale invariance allows optimizing on a sphere with a fixed weight norm [6], classic SGD-based approaches are usually preferred over constraint optimization methods in practice due to more straightforward implementation. Making an SGD step in the direction of the loss gradient always increases the norm of scale-invariant parameters, while WD aims at decreasing the weight norm (see illustration in Figure 2). In sum, training the neural network with BN and WD results in an interplay between two forces: a "centripetal force" of the WD and the "centrifugal force" of the loss gradients. Many works notice the positive effect of WD on optimization and generalization caused by the control of the scale-invariant weights norm and the subsequent influence on the *effective learning rate* [25, 10, 29, 18, 19, 26, 20], i.e., the learning rate on a unit sphere in the scale-invariant weights space. However, the general dynamics of the norm of the scale-invariant weights are viewed in the literature from two contradicting points, and this work is devoted to resolving this contradiction.

---

*First two authors contributed equally.

35th Conference on Neural Information Processing Systems (NeurIPS 2021)

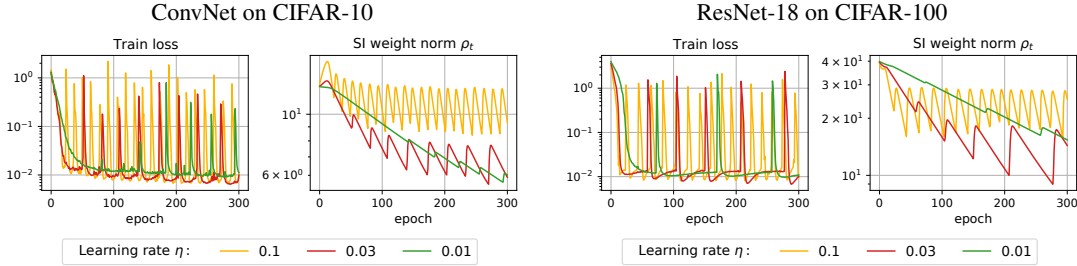

Figure 1: Periodic behavior of ConvNet on CIFAR-10 and ResNet-18 on CIFAR-100 trained using SGD with weight decay of 0.001 and different learning rates. All weights are trainable, including non-scale-invariant ones.

On the one hand, Li et al. [19] claim that learning with SGD, BN, and WD leads to an *equilibrium* state, where the "centripetal force" is compensated by the "centrifugal force" and eventually the norm of scale-invariant weights (along with other statistics related to the training procedure) will converge to a constant value. Several other works hold a similar equilibrium view [25, 5, 26]. On the other hand, a number of works [17–19] underline that using WD may cause approaching the origin (zero scale-invariant weights), which results in training *instability* due to increasing effective learning rate. Particularly, Li et al. [17] reveal that approaching the origin in weight-normalized neural networks leads to numerical overflow in gradient updates and subsequent training failure. Li and Arora [18] also underline that scale-invariant functions are ill-conditioned near the origin and prove in a simplified setting that loss convergence is impossible if both BN and WD are used (but guaranteed if either of them is disabled). Moreover, despite their equilibrium view, Li et al. [19] empirically observe that the train loss permanently exhibits oscillations between low and high values when full-batch gradient descent is used.

In this work, we study the specified contradiction between the *equilibrium* presumption and the *instability* presumption and show that both are true only to some extent. Specifically, we show that the training process converges to a consistent *periodic* behavior, i.e., it regularly exhibits instabilities which, however, do not lead to a complete training failure but cause a new period of training (see Figure 1). Thus, our contributions are as follows.

- We discover the periodic behavior of neural network training with BN and WD and reveal its reasons by analyzing the underlying mechanism for fully scale-invariant neural networks trained with standard constant learning rate SGD (Section 4) or GD (Appendix C).
- We provide a theoretical grounding for our findings by generalizing previous results on the equilibrium condition, analyzing the necessary conditions for destabilization of training, and relating the frequency of destabilization to the choice of hyperparameters (Section 5).
- We conduct a rigorous empirical study of this periodic behavior (Section 6) and show its presence in more practical scenarios with momentum, augmentation, and neural networks incorporating trainable non-scale-invariant weights (Section 7), and also with Adam optimizer and other normalization techniques (Appendix I).

Our source code is available at `https://github.com/tipt0p/periodic_behavior_bn_wd`.

## 2   Background

As discussed in the introduction, batch normalization makes (a part of) neural network's weights scale-invariant. In this section, we describe the properties of scale-invariant functions, upon which we build our further reasoning. Consider an arbitrary scale-invariant function $f(x)$, i.e., $f(\alpha x) = f(x)$, $\forall x$ and $\forall \alpha > 0$. Then two fundamental properties may be inferred, see Lemma 1.3 in Li and Arora [18]:

$$
\begin{cases}
\langle \nabla f(x), x \rangle = 0, \ \forall x & \text{(1a)} \\
\nabla f(\alpha x) = \dfrac{1}{\alpha} \nabla f(x), \ \forall x, \ \alpha > 0. & \text{(1b)}
\end{cases}
$$

Consider optimizing $f(x)$ w.r.t. $x$ using (S)GD[2] with learning rate $\eta$ and weight decay $\lambda$:

$$x_{t+1} = (1 - \eta\lambda)x_t - \eta\nabla f(x_t). \tag{2}$$

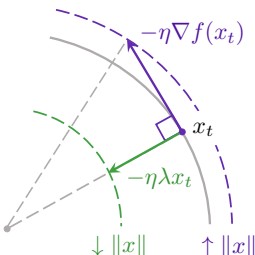

The properties above lead to two important corollaries about the dynamics of the optimization process. First, according to property (1a), shifting $x$ in the direction of $-\nabla f(x)$, i.e., making a gradient descent step, always increases $\|x\|$, while weight decay, on the other hand, decreases $\|x\|$. See Figure 2 for the illustration. The interaction of these "centripetal" and "centrifugal" forces may cause $\|x\|$ to change nontrivially during optimization. Second, according to property (1b), even though function value $f(x)$ is invariant to multiplying $x$ by $\alpha$, the optimization dynamics changes substantially when optimization is performed at different scales of $\|x\|$. For smaller norms, optimization makes larger steps, which may result in instabilities, while for larger norms, steps are smaller, and optimization process may converge slowly.

Figure 2: An illustration of the "centripetal force" of the weight decay and the "centrifugal force" of the function gradient in the optimization of scale-invariant functions.

Since scale-invariant $f(x)$ may be seen as a function on a sphere, its optimization dynamics are often analysed on a unit sphere $\|x\| = 1$. One can obtain equivalent optimization dynamics on the unit sphere as in the initial space by using the notion of *effective gradient* and *effective learning rate*. The effective gradient is defined as a gradient for a point on a unit sphere and may be obtained by substituting $\alpha = \|x\|^{-1}$ in (1b): $\nabla f(x/\|x\|) = \nabla f(x)\|x\|$. The effective learning rate can be defined as $\tilde{\eta} = \eta/\|x\|^2$ [10, 20]. Change in $\|x\|$ does not affect the effective gradient by definition and is reflected only in the effective learning rate: the lower the norm, the higher the effective learning rate, and the larger the optimization steps.

## 3 Methodology and experimental setup

In order to isolate the effect of the joint use of batch normalization and weight decay and avoid the influence of other factors, we conduct a series of experiments in a simplified setting, when all learnable weights of a neural network are scale-invariant and optimization is performed using SGD with constant learning rate, without momentum or data augmentation. This allows us to better understand the nature of the periodic behavior (Section 4) and analyse its empirical properties (Section 6). After that, we return to the setting with a more conventional training of standard neural networks and show that the periodic behavior occurs in this scenario as well (Section 7).

We conduct experiments with ResNet-18 and a simple 3-layer batch-normalized convolutional neural network (ConvNet)[3] on CIFAR-10 [15] and CIFAR-100 [16] datasets. To make standard networks fully scale-invariant, we rely on the approach of Li and Arora [18], i.e., we insert additional BN layers and fix the non-scale-invariant weights to be constant. Specifically, we use zero mean and unit variance in batch normalization layers instead of learnable location and scale parameters and freeze the weights of the last layer at random initialization. The latter action does not hurt the performance in practice [11]. However, we find that even with low train error, the training dynamics with the fixed last layer may still substantially differ from conventional training, as the neural network exhibits low confidence in predictions. To achieve high confidence for all objects and, consequently, low train loss, we increase the norm of the last layer's weights to 10. The influence of this rescaling is shown in Appendix G.

We optimize cross-entropy loss, use the batch size of 128 and train neural networks for 1000 epochs to show the consistency of the discovered periodic behavior. We consider a range of learning rates, $\{10^{-k}, 3 \cdot 10^{-k}\}_{k=0,1,2,3}$ and choose the most representative ones for each visualization, since it is difficult to distinguish many periodic functions on one plot. For fully scale-invariant neural networks, training with a fixed weight decay – learning rate product converges to similar behavior, regardless of their ratio: we show it empirically and discuss it from the theoretical point of view in Appendix F.1; the same was noticed in [18, 19]. Thus, in the main text, we provide only the results for the varied

---

[2]Since both stochastic and full-batch gradients of a scale-invariant objective possess properties (1a) and (1b), we do not distinguish between them in our reasoning.

[3]Both architectures in the implementation of `https://github.com/g-benton/hessian-eff-dim`.

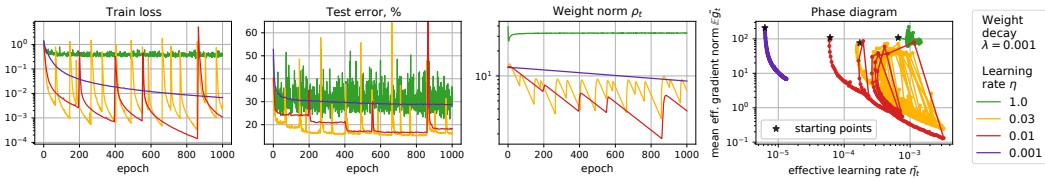

Figure 3: Periodic behavior of scale-invariant ConvNet on CIFAR-10.

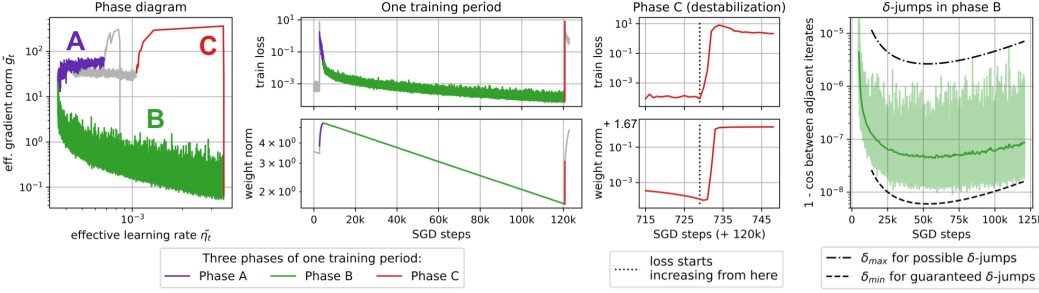

Figure 4: A closer look at one training period for scale-invariant ConvNet on CIFAR-10 trained using SGD with weight decay of 0.001 and the learning rate of 0.01. Three phases of the training period are highlighted. The train loss and the effective gradient norm computed over a mini-batch are logged after each SGD step (one epoch consists of 391 SGD steps). The rightmost plot compares empirically observed cosine distance between weights at adjacent SGD steps with theoretically derived bounds in Section 5.1. Cosine distance is presented along with the smoothed trend.

learning rate and the fixed weight decay of $0.001$. Results for the varied weight decay are presented in Appendix F.3.

At each training epoch, we log standard train / test metrics, the norm of scale-invariant weights (SI weight norm), which is in the focus of this research, and metrics characterizing training dynamics on a unit sphere: effective learning rate and the norm of effective gradients (mean over mini-batches). We plot the two latter metrics over two axes of the *phase diagram* to visualize their simultaneous dynamics that will help us to understand the mechanism underlying the periodic behavior.

## 4 Periodic behavior and its underlying mechanism

As discussed in the previous section, we begin our study by considering a simplified setting with a fully scale-invariant neural network trained with standard SGD. Figure 3 shows the presence of the periodic behavior for a scale-invariant ConvNet on the CIFAR-10 dataset for a range of learning rates. In Appendix F.2, we show the presence of the periodic behavior for other dataset-architecture pairs. The same periodic behavior is also present for neural network training with full-batch gradient descent, see Appendix C. Moreover, this behavior can be observed even when optimizing common scale-invariant functions using the gradient descent method with weight decay (see Appendix E).

The observed periodic behavior occurs because of the interaction between batch normalization and weight decay, particularly because of their competing influence on the weight norm. As discussed in Section 2, weight decay aims at decreasing the weight norm, while loss gradients aim at increasing the weight norm due to scale invariance caused by batch normalization (see Figure 2). These two forces alternately outweigh each other for quite long periods of training, resulting in periodic behavior.

Let us examine a single period in greater detail by analyzing Figure 4 that shows the dynamics of relevant training metrics logged after each SGD step of ConvNet training. At the beginning of the period, the train loss is high, and the large gradients of the loss outweigh weight decay. This results in increasing weight norm and decreasing effective learning rate, i.e., we move along phase $A$ of the phase diagram. SGD continues optimizing train loss, and at some point, train loss and its gradients become small and outweighed by weight decay. As a result, the weight norm starts decreasing, and the effective learning rate increases, i.e., we move along phase $B$ of the phase diagram. We note

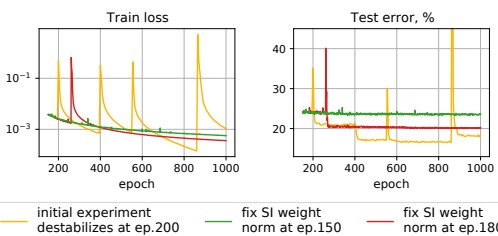

Figure 5: The absence of the periodic behavior for training with the fixed weight norm. Scale-invariant ConvNet on CIFAR-10 trained using SGD with weight decay of 0.001 and learning rate of 0.01. Left pair: the weight norm is fixed at random initialization of different scales. Right pair: the weight norm is fixed at some epoch of regular training before destabilization.

that the transition between phases $A$ and $B$ correlates with achieving near-zero train error. When the weight norm becomes too small, and the effective learning rate becomes too high, SGD makes several large steps and leaves the low loss region. Gradients grow along with train loss and, multiplied by a high effective learning rate, lead to the fast growth of the weight norm, i.e., we move along phase $C$ of the phase diagram. The detailed plot of phase $C$ in Figure 4 confirms that train loss starts increasing earlier than the weight norm. When the weight norm becomes large, the effective learning rate becomes low and stops the process of divergence. After that, a new period of training begins.

We also conducted an ablation experiment to show that the discovered periodic behavior is indeed a result of the competing influence of BN and WD on the weight norm. To do so, we prohibit this influence and train the network on a sphere by fixing the weight norm and rescaling the weights after each SGD step. We firstly fix the weight norm at random initialization, considering different values of the initialization weight norm and hence different (fixed) effective learning rates. Figure 5 (left pair) shows that in this case, there is no periodic behavior, and the train loss either converges (for relatively low effective learning rates) or gets stuck at high values (for high effective learning rates). We repeat this ablation fixing the weight norm at some epoch preceding destabilization in the experiment where we observe the periodic behavior. Specifically, as an initial experiment, we use the one with the learning rate of 0.01 from Figure 3 and fix the weight norm at the 150-th and 180-th epochs, preceding the destabilization at epoch 200. Figure 5 (right pair) shows the absence of the periodic behavior in both cases. When we fix the weight norm closer to the destabilization at the 180-th epoch, we observe a single increase in train loss, as the training process has already become unstable. However, after converging from this increase, train loss never destabilizes again.

## 5 Theoretical grounding for periodic behavior

In this section, we theoretically investigate the reasons for the training destabilization between phases $B$ and $C$, and after that, we generalize the overall training process equilibrium condition of Li et al. [19] taking into account the discovered periodic behavior. To do so, we study the optimization dynamics of an arbitrary scale-invariant function $f(x)$ trained using (S)GD with learning rate $\eta$ and weight decay of strength $\lambda$ (2). Hereinafter, we will assume that the $\eta\lambda$ product is small, i.e., we can suppress $\mathcal{O}\left((\eta\lambda)^2\right)$ terms. We also refer to Appendix A for the proofs, derivations, and further discussion on our theoretical results.

We recall that (stochastic) gradients of an arbitrary scale-invariant function $f(x)$ possess two fundamental properties (1a) and (1b). Based on these properties, we obtain the dynamics of the parameters norm induced by Eq. (2) which we also leverage in our analysis (derivation of this and other equations is deferred to Appendix A.2):

$$\rho_{t+1}^2 = (1 - \eta\lambda)^2 \rho_t^2 + \eta^2 \tilde{g}_t^2 / \rho_t^2, \qquad (3)$$

where $\rho_t = \|x_t\|$ denotes the parameters norm, $g_t = \|\nabla f(x_t)\|$ — the gradient norm, $\tilde{g}_t = \|\nabla f(x_t / \|x_t\|)\| = \rho_t g_t$ — the effective gradient norm. In this work, we also use the notion of effective learning rate which is formally defined as $\tilde{\eta}_t = \eta / \rho_t^2$.

## 5.1 The notion of $\delta$-jumps

As scale-invariant functions are essentially defined on a sphere, cosine distance is a natural choice for a metric in parameter space. The following notion defines a situation when adjacent iterates become distant from each other, indicating training destabilization.

**Definition 1** *We say that dynamics* (2) *performed a **$\delta$-jump** once the cosine distance between adjacent iterates exceeds some value $\delta > 0$:*

$$1 - \cos(x_t, x_{t+1}) > \delta.$$

We conjecture that *the necessary condition for the training dynamics' destabilization is performing $\delta$-jumps with sensible values of $\delta$*. Otherwise, as long as adjacent iterates remain too close, the model (and hence its training dynamics) cannot change significantly. This holds strictly, for instance, if the Lipschitz constant of $f$ is bounded (at least locally), which can be relevant for neural networks with BN [22]. But even if $f$ has unstable regions on a unit sphere with very high or even unbounded Lipschitz constant, our analysis is still relevant since making larger steps in such regions would lead to a higher chance of divergence. Further, we show that the closer we approach the origin, the larger effective steps (steps on a unit sphere) we start making, thereby paving the way for destabilization.

Now, our question is, *given the value $\delta$, what are the conditions for a $\delta$-jump to occur?* By assuming that effective gradients are bounded, i.e., we can set two values $0 \leq \ell \leq L < +\infty$ such that $\tilde{g}_t \in [\ell, L]$, we answer this question in the following proposition. Proof can be seen in Appendix A.3.

**Proposition 1** *If $f(x)$ is a scale-invariant function optimized according to dynamics* (2) *with bounded effective gradients $0 \leq \ell \leq \tilde{g}_t \leq L < +\infty$, then for sufficiently small $\delta$ and assuming $(1 - \eta\lambda) \lessapprox 1$, the following approximate conditions on $\delta$-jump hold:*

$$\begin{cases} \rho_t^2 \lessapprox \dfrac{\eta L}{\sqrt{2\delta}} \implies \delta\text{-jump is possible}, & \text{(4a)} \\[3mm] \rho_t^2 \lessapprox \dfrac{\eta \ell}{\sqrt{2\delta}} \implies \delta\text{-jump is guaranteed}. & \text{(4b)} \end{cases}$$

**Remark 1** *Our results hold for any values $\ell$, $L$ bounding the effective gradient norm, but the tighter these bounds are, the more precisely our theory describes the properties of the actual dynamics, thus we generally assume that $\ell$ and $L$ are taken as local bounds on $\tilde{g}_t$ valid for several current iterations.*

To connect our theoretical results with practice, we examine the behavior of effective steps length of a scale-invariant neural network, compare it with theoretical bounds and observe gradually increasing destabilization of training dynamics. The rightmost plot of Figure 4 visualizes the cosine distance $1 - \cos(x_t, x_{t+1})$ between neural network's weights $x_t$ and $x_{t+1}$ at adjacent SGD steps for phase $B$ of the training period. The dashed lines denote the theoretical upper and lower bounds on the cosine distance corresponding to the maximal and minimal $\delta$-jumps derived from Eq. (4a) and (4b), respectively: $\delta_{\max} = \frac{\eta^2 L^2}{2\rho_t^4}$, $\delta_{\min} = \frac{\eta^2 \ell^2}{2\rho_t^4}$. To obtain those, we calculated the network's parameters norm $\rho_t$ at each iteration and chose $\ell$ and $L$ as smooth functions locally bounding the effective gradient norm in phase $B$ (see Appendix D for details). We can see that both bounds, along with the measured cosine distance, start growing in the second half of the phase. This indicates that the performed $\delta$-jumps are gradually increasing, hence instability accumulates until the training diverges. We note that such a long-lasting increase in cosine distance is common but, in general, not obligatory in the case of training with SGD because SGD may exhibit destabilization even with small $\delta$-jumps due to stochasticity. For full-batch GD, this effect is even more prominent, see Appendix C.

Next, we formulate a proposition about how the initial parameters norm value $\rho_0$ and hyperparameters $\eta$ and $\lambda$ affect the time of $\delta$-jumps occurrence and hence the frequency of the periods since training dynamics destabilization is closely connected with $\delta$-jumps. Proof can be found in Appendix A.5. Note that $\rho_0$ should be interpreted as the norm at some initial moment $t = 0$ of a given period, when the conditions of the proposition are met, (typically, at the beginning of phase B) rather than the norm after initialization, i.e., at the very first iteration of training.

**Proposition 2** *Denote $\kappa = \sqrt{\frac{\eta}{2\lambda}}$. Under the assumptions of Proposition 1:*

1. *if $\rho_0^2 > \kappa\ell$ and $\delta < \eta\lambda\frac{L^2}{\ell^2}$, then the **minimal** time required for the $\delta$-jump to occur:*

$$t_{\min} = \max\left\{0, \frac{\log\left(\rho_0^2 - \kappa\ell\right) - \log\left(\frac{\eta L}{\sqrt{2\delta}} - \kappa\ell\right)}{-\log(1 - 4\eta\lambda)}\right\};$$ (5)

2. *if $\rho_0^2 > \kappa L$ and $\delta < \eta\lambda\frac{\ell^2}{L^2}$, then the **maximal** time required for the $\delta$-jump to occur:*

$$t_{\max} = \max\left\{0, \frac{\log\left(\rho_0^2 - \kappa L\right) - \log\left(\frac{\eta\ell}{\sqrt{2\delta}} - \kappa L\right)}{-\log(1 - 2\eta\lambda)}\right\}.$$ (6)

**Corollary 1** *Since both $t_{\max}$ and $t_{\min}$ are inversely proportional to $\eta\lambda$ as $-\log(1 - \varepsilon) \approx \varepsilon$ for small $\varepsilon$, $\delta$-jumps (and hence periods) must occur more often for larger values of $\eta\lambda$.*

### 5.2 Generalization of the equilibrium condition

We now generalize the equilibrium condition of Li et al. [19] and characterize the behavior of the parameters norm globally in the following proposition. The proof is provided in Appendix A.6.

**Proposition 3** *Denote $\kappa = \sqrt{\frac{\eta}{2\lambda}}$. Under the assumptions of Proposition 1, if $2\eta\lambda L \leq \ell$, then*

$$\kappa\ell \leq \rho_t^2 \leq \kappa L, \, t \gg 1.$$ (7)

*Furthermore, if $\rho_0^2 > \kappa L$, then $\rho_t^2$ converges linearly to $[\kappa\ell, \kappa L]$ interval in $\mathcal{O}\left(1/\eta\lambda\right)$ time.*

Note that Li et al. [19] and Wan et al. [26] similarly predict that the equilibrium state can be reached in a linear rate regime. The condition $2\eta\lambda L \leq \ell$ is generally fulfilled in practice for small $\eta\lambda$ product even for globally chosen bounds $\ell, L$. A similar assumption is made, e.g., in Theorem 1 in Wan et al. [26]. We discuss it in more detail (including the non-fulfillment case) in Appendix A.7.

Proposition 3 generalizes the results of Li et al. [19] who claimed that the effective learning rate $\tilde{\eta}_t = \eta/\rho_t^2$ converges to a constant. Their derivation relies on the assumption of stabilization of the effective gradient variance, which contradicts the observed periodic behavior. We relax this assumption by putting bounds on the effective gradient norm, thus bounding the parameters norm limits. These bounds can be either local, which defines the local trend of parameters norm dynamics, or global, which describes its general behavior. Also, note that we, in some sense, extend the results of Wan et al. [26] as we provide the exact limiting interval for $\rho_t^2$, not just bound its variance.

## 6 Empirical study of the periodic behavior

After discussing the reasons for the occurrence of the periodic behavior, we now further analyze its properties. In particular, we investigate: how hyperparameters affect the periodic behavior, how the periodic behavior evolves over epochs, and how minima achieved in different training periods differ both in parameter and functional space. In this section, we again consider the simplified setting with a fully scale-invariant neural network trained with standard SGD.

**Influence of hyperparameters.** We investigate the influence of two key training hyperparameters: learning rate and weight decay, but since the training dynamics mainly depend on their product (see discussion in Section 3 and Appendix F.1), we only vary the learning rate. The results for ConvNet on CIFAR-10 are given in Figure 3, the results for other dataset-architecture pairs are presented in Appendix F.2, and the results on the variable weight decay are given in Appendix F.3. Our first observation is that with higher learning rates, consistent periodic behavior occurs at larger weight norms. This is because SGD with a high learning rate can only converge with relatively small gradient norms, which are achieved at large weight norms according to Eq. (1b). This observation also agrees with Proposition 3 in Section 5. The second observation is that the periodic behavior is present for a wide range of learning rates, e.g., $0.003 - 0.3$ for ConvNet on CIFAR-10, and the higher the learning

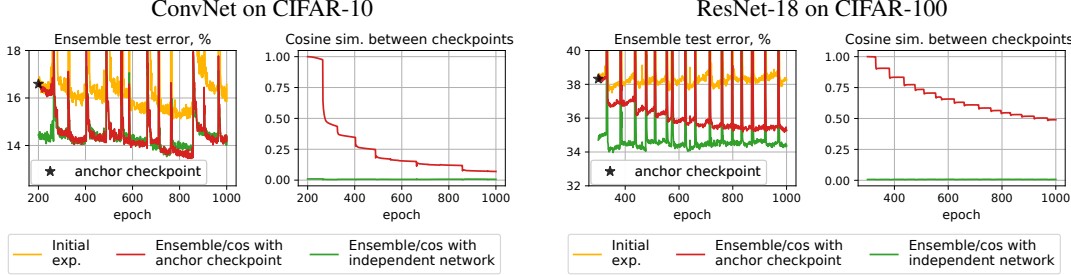

Figure 6: Similarity in the weight space (cosine sim.) and the functional space (ensemble test error) for different checkpoints of training scale-invariant ConvNet on CIFAR-10 (left pair) and ResNet on CIFAR-100 (right pair) using SGD with weight decay of 0.001 and learning rate of 0.03.

rate, the shorter the periods, which agrees with Corollary 1 in Section 5. When using a learning rate that is too high, e.g., 1 in Figure 3, we expect training to yield very large weight norms, however weight decay prohibits us from reaching them, thus the gradients are not able to shrink sufficiently, and training gets stuck at high train loss. On the other hand, using a learning rate that is too low, e.g., 0.001 in Figure 3, leads to prolonged training which does not reach a small enough weight norm to yield a high effective learning rate, resulting in the absence of the periodic behavior in the given number of epochs. We note that the periodic behavior is present for the learning rates giving the lowest test error. In Appendix F.3, we show that varying weight decay leads to similar effects: the periodic behavior is present for a wide range of reasonable weight decays but is absent for too low or too high weight decays, and the higher the weight decay, the faster the periods.

**Dynamics of periodic behavior.**    We now analyze how the discovered periodic behavior evolves over epochs. As discussed in the previous paragraph, consistent periodic behavior occurs at larger weight norms with higher learning rates. However, the initialization may have a substantially different norm. Thus we observe a *warm-up stage* in some plots of Figures 1 and 3, when the beginning of training is spent on moving towards the appropriate norm of scale-invariant weights. Expectedly, this warm-up stage is more prolonged for lower learning rates. Reaching the proper weight norm initiates a consistent periodic behavior. During the warm-up stage, SGD can still exhibit regular destabilization happening at higher weight norms than in the stage of consistent periodic behavior. We hypothesize that at the early stage of training, SGD converges to less stable basins with larger effective gradients, in which destabilization happens at larger norms of the scale-invariant parameters. We notice that test error decreases after each warm-up destabilization episode and reaches a lower level than training with a fixed effective learning rate, as shown in Figure 5 (right pair). In other words, the performance may benefit from the repeating destabilization.

**Minima achieved at different training periods.**    Next, we aim at understanding whether minima achieved in different training periods are close in weight and functional spaces. We use the cosine similarity function for the weight space and estimate similarity in the functional space by comparing with ensembles of independent models, following Fort et al. [8]. If training process converges to the same minimum in each period, then cosine similarity between two minima achieved in different periods should be close to one and their ensemble error should be close to the error of a single network. On the contrary, if destabilization moves training so far that it is equivalent to retraining a model from a new random initialization, then the cosine similarity between the two minima should be close to zero and their ensemble error should be close to the error of an ensemble of two independently trained networks.

The setup of the experiment is as follows. We select some initial experiment and its checkpoint (called anchor checkpoint) corresponding to the minimum achieved when the training process has already converged to the consistent periodic behavior. After that, we measure weight/function similarities between the anchor checkpoint and all the subsequent checkpoints of the initial experiment — this is our primary measurement. For comparison, we independently train one more neural network with the same hyperparameters as in the initial experiment but from a different random initialization, select its checkpoint with the same test error as that of the anchor checkpoint, and measure the similarity between this new checkpoint and the checkpoints of the original experiment.

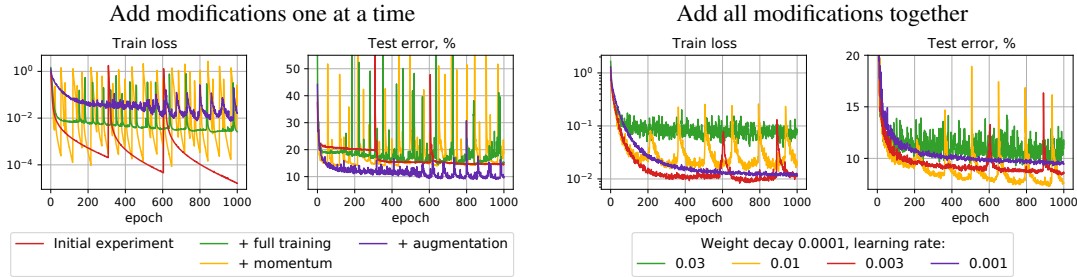

Figure 7: Periodic behavior of ConvNet (of increased width) on CIFAR-10 trained with the more practical modifications. Left: weight decay of 0.001, learning rate of 0.01.

The results for ConvNet on CIFAR-10 and ResNet-18 on CIFAR-100 are presented in Figure 6, the results for other dataset-architecture pairs are given in Appendix H. Inside one training period, checkpoints do not step far from the anchor checkpoint, i.e., the cosine similarity is close to one, and the ensemble test error is close to the error of a single network. However, when the next training period begins after destabilization, SGD moves to another region in the weight space, and both similarities start decreasing: the cosine similarity drops, and ensemble test error becomes smaller than that of a single network. Each following training period moves SGD farther away from the anchor checkpoint. For networks on CIFAR-100, late training periods continue to be correlated with the anchor checkpoint, i.e., the cosine similarity only reaches $\sim 0.5$ value, and ensemble test error does not reach the level of the independent networks ensemble. Still, both similarities continue decreasing. For networks on CIFAR-10, the cosine similarity decreases faster, and the ensemble test error quickly reaches the test error of an ensemble of two independently trained networks. To sum up, minima achieved at two neighboring training periods are substantially different, but their similarity is usually higher than that of two independently trained networks.

## 7 Periodic behavior in a practical setting

In the previous sections, we conducted experiments with scale-invariant neural networks trained with the simplest version of SGD. This allowed us to analyze the periodic behavior of train loss in detail. However, in practice, a portion of the weights of a neural network are not scale-invariant, e.g., the weights of the last layer and learnable BN parameters. Furthermore, networks are trained using more advanced procedures, e.g., SGD with momentum, data augmentation, and a learning rate schedule. At the same time, periodic behavior was mainly not noticed in previous works to the best of our knowledge. In this section, we show the presence of the periodic behavior for standard neural networks trained with momentum and data augmentation and discuss why periodic behavior may be not observed in practice. In Appendix I, we also show the presence of the periodic behavior for the networks with other normalization approaches or trained with Adam [13].

We select one of our initial experiments and add modifications one at a time to see their effects more clearly. We also present the results for training with all modifications turned on together. The plots for ConvNet on CIFAR-10 are given in Figure 7 and for other setups — in Appendix I. In this section, we use a wider ConvNet, as the standard version is too small to learn the augmented dataset.

**Training non-scale-invariant weights.** To achieve full scale-invariance, we froze the weights of the last layer and the parameters of BN layers since they all are not scale-invariant. We now consider the conventional procedure that implies training all neural network weights. In addition to the results presented in Figure 7 (left), we refer the reader to Figure 1. We observe that training non-scale-invariant weights retains the periodic behavior and affects the frequency of periods. The last-mentioned effect relates to the trainable last layer that automatically adjusts prediction confidence. In Appendix G, we show that variable prediction confidence results in different periodic behavior.

**SGD with momentum.** Next, we investigate the effect of using a more complex optimization algorithm. We consider SGD with momentum as the algorithm most commonly used for training convolutional neural networks. We observe that using momentum does not break the periodic

behavior and increases the frequency of periods. This agrees with the commonly observed effect that momentum speeds up training [23], i.e., momentum accelerates phases $A$ and $B$. Interestingly, momentum does not prevent destabilization.

**Data augmentation.** We next consider training on the dataset with standard CIFAR-10 data augmentations, see details in Appendix B. Augmentation prevents over-fitting to the training data, which results in less confident predictions and larger train loss. As a result, train loss gradients outweigh WD more easily. If the number of parameters in the neural network is insufficient to achieve low train loss gradients, phase $A$ never ends (at least in 1000 epochs), resulting in the absence of the periodic behavior. On the other hand, a sufficiently large neural network learns the augmented dataset at some epoch and proceeds to phase $B$, launching the periodic process. Still, we note that the periodic behavior begins much later than for the network trained without augmentation. This is one of the main reasons why the periodic behavior is often not observed in practice: it requires a much larger number of epochs to start than conventionally used for training.

**All modifications together.** In the two right plots of Figure 7, we visualize training with momentum, data augmentation, and unfrozen non-scale-invariant parameters used simultaneously and observe the presence of the periodic behavior.

So, what factors do interfere with observing the periodic behavior in practice? We underline two main factors. First, the interplay between different modifications narrows the range of hyperparameter values for which periodic behavior is present. When non-scale-invariant parameters are trained, the model converges to low test error only with specific values of weight decay. Moreover, with data augmentation, periodic behavior occurs only with relatively high learning rates (with lower learning rates, the training is too slow to reach phase $C$ in 1000 epochs), while with momentum, using too high learning rates may result in training failure in phase $A$. In sum, periodic behavior appears only for a limited range of hyperparameters. Despite that, we note that the model generally achieves its best performance exactly in this range. Second, practical settings also imply learning rate schedules and a relatively small number of epochs, which do not preserve periodic behavior. We provide further discussion on comparison of our experimental setup with other works in Appendix J.

# 8 Conclusion

In this work, we described the periodic behavior of neural network training with batch normalization and weight decay occuring due to their competing influence on the norm of the scale-invariant weights. The discovered periodic behavior clarifies the contradiction between the equilibrium and instability presumptions regarding training with BN and WD and generalizes both points of view. In our empirical study, we investigated what factors and in what fashion influence the discovered periodic behavior. In our theoretical study, we introduced the notion of $\delta$-jumps to describe training destabilization, the cornerstone of the periodic behavior, and generalized the equilibrium conditions in a way that better describes the empirical observations.

**Limitations and negative societal impact.** We discuss only conventional training of convolutional neural networks for image classification and do not consider other architectures and tasks. However, we believe that our findings extrapolate to training any kind of neural network with some type of normalization and weight decay. We also focus on a particular source of instability induced by BN and WD, yet, other factors may make training unstable [7]. This is an exciting direction for future research. To the best of our knowledge, our work does not have any direct negative societal impact. However, while conducting the study, we had to spend many GPU hours, which, unfortunately, could negatively affect the environment.

## Acknowledgments and Disclosure of Funding

We would like to thank Sofia Sibagatova for the help with Appendix I. The work was supported in part by the Russian Science Foundation grant №19-71-30020. The empirical results were supported in part through the computational resources of HPC facilities at HSE University [14]. Additional revenues of the authors for the last three years: laboratory sponsorship by Samsung Research, Samsung Electronics; travel support by Google, NTNU, DESY, UCM.

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
