# A  Theoretical results

This section contains details on our theoretical results.

## A.1  Invariance to hyperparameters rescaling

Based on properties (1a) and (1b), we derive a simple yet useful proposition tying together different hyperparameter settings of initialization $x_0$, learning rate $\eta$, and weight decay coefficient $\lambda$. This proposition provides grounds for fixing the initialization scale in our experiments and iterating over learning rates and weight decay coefficients when studying the dependence of the behavior of scale-invariant neural networks on hyperparameters.

**Proposition 4** *Given $f(x)$ is scale-invariant and optimized according to Eq. (2), settings $(x_0, \eta, \lambda)$ and $(x_0', \eta', \lambda') = (cx_0, c^2\eta, \lambda/c^2)$, $c > 0$ lead to equivalent dynamics in function space.*

**Proof.** Eq. (2) and property (1b) give $x_{t+1} = \|x_t\| \left[ (1 - \eta\lambda)\frac{x_t}{\|x_t\|} - \tilde{\eta}_t \nabla f(x_t / \|x_t\|) \right]$, where $\tilde{\eta}_t = \frac{\eta}{\|x_t\|^2}$ is the effective learning rate. Since the term in square brackets does not depend on the scale of $x_t$ provided that the effective learning rate and $\eta\lambda$ product are unchanged, by induction, from $x_t' = cx_t$ we have $x_{t+1}' = cx_{t+1}$, hence $f(x_{t+1}') = f(x_{t+1})$. ∎

## A.2  Derivations

**Parameters norm dynamics** (3)
$$\rho_{t+1}^2 = \langle x_{t+1}, x_{t+1} \rangle = \{\text{Eq. (2)}\} = (1 - \eta\lambda)^2 \rho_t^2 + \eta^2 g_t^2 + 2\eta(1 - \eta\lambda) \langle \nabla f(x_t), x_t \rangle =$$
$$= \{\text{property (1a)}\} = (1 - \eta\lambda)^2 \rho_t^2 + \eta^2 g_t^2 = \{\text{property (1b), i.e., } g_t = \tilde{g}_t/\rho_t\} =$$
$$= (1 - \eta\lambda)^2 \rho_t^2 + \eta^2 \tilde{g}_t^2 / \rho_t^2$$

**Cosine distance between adjacent iterates** (8)
$$\cos(x_t, x_{t+1}) = \frac{\langle x_t, x_{t+1} \rangle}{\rho_t \rho_{t+1}} = \{\text{Eq. (2)}\} = \frac{(1 - \eta\lambda) \langle x_t, x_t \rangle - \eta \langle \nabla f(x_t), x_t \rangle}{\rho_t \rho_{t+1}} =$$
$$= \{\text{property (1a)}\} = \frac{(1 - \eta\lambda)\rho_t}{\rho_{t+1}} = \{\text{Eq. (3)}\} = \left( 1 + \frac{\eta^2 \tilde{g}_t^2}{(1 - \eta\lambda)^2 \rho_t^4} \right)^{-1/2}$$

**$\delta$-jump conditions** (9)

$$1 - \cos(x_t, x_{t+1}) > \delta \iff \{\text{Eq. (8)}\} \iff \left( 1 + \frac{\eta^2 \tilde{g}_t^2}{(1 - \eta\lambda)^2 \rho_t^4} \right)^{-1/2} < 1 - \delta \iff$$
$$\iff 1 + \frac{\eta^2 \tilde{g}_t^2}{(1 - \eta\lambda)^2 \rho_t^4} > \frac{1}{(1 - \delta)^2} = 1 + 2\delta + \mathcal{O}(\delta^2) \gtrsim 1 + 2\delta.$$

Omitting $\mathcal{O}(\delta^2)$ leaves the condition necessary and also approximately sufficient for small $\delta$:

$$1 - \cos(x_t, x_{t+1}) > \delta \implies \frac{\eta^2 \tilde{g}_t^2}{(1 - \eta\lambda)^2 \rho_t^4} > 2\delta \iff \rho_t^2 < \frac{\eta \tilde{g}_t}{(1 - \eta\lambda)\sqrt{2\delta}}.$$

## A.3  Proof of Proposition 1

For the convenience of reading, we defer the derivation details of all equations to Appendix A.2.

**Proof.** Using property (1a) and Eq. (3), we obtain the exact value of the cosine between adjacent iterates:

$$\cos(x_t, x_{t+1}) = \left( 1 + \frac{\eta^2 \tilde{g}_t^2}{(1 - \eta\lambda)^2 \rho_t^4} \right)^{-1/2}. \tag{8}$$

From Eq. (8) we deduce the following $\delta$-jump condition:

$$1 - \cos(x_t, x_{t+1}) > \delta \implies \rho_t^2 < \frac{\eta \tilde{g}_t}{(1 - \eta\lambda)\sqrt{2\delta}}. \tag{9}$$

During the derivation, we omitted $\mathcal{O}(\delta^2)$ terms. This implies that the right inequality represents not only the necessary but also (approximately) the sufficient condition for a $\delta$-jump when $\delta$ is small.

Assuming $(1 - \eta\lambda) \lesssim 1$ and substituting the effective gradient bounds $\ell$ and $L$ into Eq. (9) in place of $\tilde{g}_t$ finally yields the approximate necessary and sufficient $\delta$-jump conditions (4a) and (4b), respectively. ∎

## A.4 On $\beta$-undetermined recurrent sequences

Here we provide some results related to convergence of sequences of the following kind:

$$x_{t+1} = (1 - \alpha)x_t + \frac{\beta_t}{x_t}, \tag{10}$$

where $\alpha$ is a fixed coefficient, and $\beta_t$ may vary from iteration to iteration. We assume $x_0 > 0$, $0 < \alpha < 0.5$, and $\beta_t \in [a, b]$, $\forall t$, where $0 \le a \le b < +\infty$ are some fixed values. We call sequences of type (10) $\beta$-*undetermined* recurrent sequences.

### A.4.1 $\beta$-determined sequences

To derive the basic properties of $\beta$-undetermined sequences (10), we first consider $\beta$-*determined* recurrent sequences:

$$x_{t+1} = (1 - \alpha)x_t + \frac{\beta}{x_t}, \tag{11}$$

where $\beta$ is now a fixed non-negative value.

If $\beta = 0$, (11) boils down to a classical linear sequence converging to zero at rate $1 - \alpha$. Assume now that $\beta > 0$. First of all, $x^* = \sqrt{\frac{\beta}{\alpha}}$ is the only stationary point of sequence (11). This holds from solving the following equation:

$$x_{t+1} = x_t \iff x_t = x^* = \sqrt{\frac{\beta}{\alpha}}.$$

Suppose $x_t = \gamma_t x^*$. By dividing the left and right sides of Eq. (11) by $x^*$, we can derive the formula for $\gamma_{t+1}$ as a function of $\gamma_t$ which we denote as $\varphi(\gamma_t)$:

$$\gamma_{t+1} = \varphi(\gamma_t) = (1 - \alpha)\gamma_t + \frac{\alpha}{\gamma_t}. \tag{12}$$

The sequence induced by (12) is a special case of Eq. (11) with a stationary point $\gamma^* = 1$. One important property is that $\gamma_{t+1}$ does not depend on $\beta$ explicitly, only on $\gamma_t$ and $\alpha$. This unifies the convergence analysis for sequences with different $\beta$ coefficients.

For function (12) the following facts hold (see Figure 8 for an illustration):

$$
\begin{cases}
\gamma_t < 1 \implies \gamma_{t+1} > \gamma_t: \text{ the sequence is increasing once it's below } x^*, & \text{(13a)} \\[1mm]
\gamma_t > 1 \implies 1 < \gamma_{t+1} < \gamma_t: \text{ the sequence is decreasing once it's above } x^*, & \text{(13b)} \\[1mm]
\gamma_{t+1} = 1 \iff \gamma_t = \dfrac{\alpha}{1 - \alpha} \vee \gamma_t = 1: \text{ pre-stationary conditions}, & \text{(13c)} \\[1mm]
\gamma_{t+1} < 1 \iff \gamma_t \in \left( \dfrac{\alpha}{1 - \alpha}, 1 \right): \text{ conditions for staying below the stationary point}, & \text{(13d)} \\[1mm]
\varphi(\gamma_t) \text{ is a decreasing function for } \gamma_t < \sqrt{\dfrac{\alpha}{1 - \alpha}}, & \text{(13e)} \\[1mm]
\varphi(\gamma_t) \text{ is an increasing function for } \gamma_t > \sqrt{\dfrac{\alpha}{1 - \alpha}}, & \text{(13f)} \\[1mm]
\gamma_{t+1} = \min_{\gamma_t} \varphi(\gamma_t) = 2\sqrt{\alpha(1 - \alpha)} \iff \gamma_t = \sqrt{\dfrac{\alpha}{1 - \alpha}}: \text{ the lowest achievable value.} & \text{(13g)}
\end{cases}
$$

Note that for $0 < \alpha < 0.5$ we have

$$\frac{\alpha}{1 - \alpha} < \sqrt{\frac{\alpha}{1 - \alpha}} < 2\sqrt{\alpha(1 - \alpha)} < 1.$$

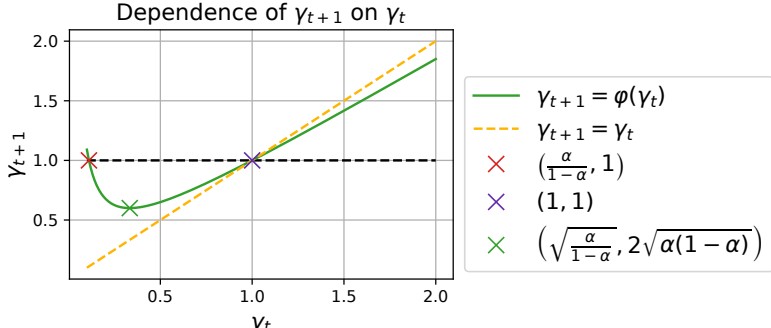

Figure 8: Dependence of $\gamma_{t+1}$ on $\gamma_t$ from Eq. (12) for $\alpha = 0.1$.

Properties (13b) and (13d) imply that $x_{t+1}$ can "hop" over $x^*$ if only $x_t < \frac{\alpha}{1-\alpha}x^*$. Otherwise, $x_t$ is monotonically approaching its stationary point. That is an important threshold that will help derive the convergence of $\beta$-undetermined sequences to a specific equilibrium interval.

The derivative of $\varphi(\gamma_t)$ can help estimate the convergence rate of the sequence (11) to its stationary point. Specifically, using the mean value theorem, we obtain that

$$x_{t+1} - x^* = x^* (\gamma_{t+1} - 1) = x^* (\varphi(\gamma_t) - \varphi(1)) = x^* \varphi'(\xi) (\gamma_t - 1), \tag{14}$$

where $\xi$ is some point between 1 and $\gamma_t$. Therefore, by bounding the derivative $\varphi'(\gamma_t)$, we can also bound the $x_t$ convergence to $x^*$.

Suppose that $\gamma_0 > 1$. From (13b) it follows that $\gamma_t > 1$, $\forall t$. In this case, we can bound the derivative of $\varphi(\gamma_t)$ for $\gamma_t > 1$ and obtain the approximate convergence rates for (11):

$$1 - 2\alpha < \varphi'(\gamma_t) = (1 - \alpha) - \frac{\alpha}{\gamma_t^2} < 1 - \alpha, \ \gamma_t > 1,$$

which, after recursively applying (14), yields

$$(1 - 2\alpha)^t(\gamma_0 - 1) < \gamma_t - 1 < (1 - \alpha)^t(\gamma_0 - 1), \ t \geq 1,$$

or equivalently, formulating this for (11) as a lemma:

**Lemma 1** *For an arbitrary $\beta$-determined sequence (11) with $\beta \geq 0$, given $x^* = \sqrt{\frac{\beta}{\alpha}}$ and $x_0 > x^*$, the following bounds on its convergence rate hold:*

$$(1 - 2\alpha)^t(x_0 - x^*) \leq x_t - x^* \leq (1 - \alpha)^t(x_0 - x^*), \ \forall t.$$

This is the main result concerning the convergence of $\beta$-determined sequences (11). Note that Lemma 1 also covers the case of $\beta = 0$ because then $x^* = 0$ and $x_t = (1 - \alpha)^t x_0$.

### A.4.2 $\beta$-undetermined sequences convergence bounds

Now, we can return back to the $\beta$-undetermined sequences (10) and derive its convergence bounds. The following lemma allows to bound an arbitrary $\beta$-undetermined sequence with $\beta$-determined ones.

**Lemma 2** *For an arbitrary $\beta$-undetermined sequence of type (10) with $0 \leq a \leq \beta_t \leq b < +\infty$ the following $\beta$-determined bounds hold.*

1. *Let $x_{a,t}$ be a $\beta$-determined sequence (11) with $\beta = a$ and $x_{a,0} = x_0$. Then $x_{a,t} \leq x_t$, $\forall t$.*

2. *Let $x_{b,t}$ be a $\beta$-determined sequence (11) with $\beta = b$ and $x_{b,0} = x_0$. Then, if $x_t > \sqrt{\frac{b}{1-\alpha}}$, $t = 0, \ldots, T$, we have $x_t \leq x_{b,t}$, $t = 0, \ldots, T + 1$.*

**Proof.** We will prove the first statement since the second one can be proved similarly.

Let $\sqrt{\frac{a}{1-\alpha}} < x_{a,t} \leq x_t$. Then the following inequalities hold:

$$x_{a,t+1} \leq (1-\alpha)x_t + \frac{a}{x_t} \leq x_{t+1}.$$

The first inequality holds since $x_{a,t+1}$ is a monotonically increasing function of $x_{a,t}$ due to (13f). The second one is valid because $a \leq \beta_t$.

Note that due to (13g) and $\sqrt{\frac{\alpha}{1-\alpha}} < 2\sqrt{\alpha(1-\alpha)}$, we have $\sqrt{\frac{a}{1-\alpha}} < x_{a,t}$, $t \geq 1$, plus, as $a \leq \beta_0$, $x_{a,1} \leq x_1$, hence, induction is valid for all $t$ for the lower bound (in contrast with the upper bound case, where we explicitly demand $x_t > \sqrt{\frac{b}{1-\alpha}}$ for $T$ consecutive timesteps). ∎

**Remark 2** *An important special case when the upper bound $x_{b,t}$ is valid for all $t$ is if $\frac{\alpha}{1-\alpha}\sqrt{b} \leq \sqrt{a}$ and $x_0 > \sqrt{\frac{b}{\alpha}}$. Then, while $x_t \geq \sqrt{\frac{b}{\alpha}} > \sqrt{\frac{b}{1-\alpha}}$ the bound is valid due to the second statement of the lemma. As soon as $x_t$ crosses the $\sqrt{\frac{b}{\alpha}}$ threshold, it can never "hop" over it again due to (13d) and $x_t \geq x_{a,t} > \sqrt{\frac{a}{\alpha}} \geq \frac{\sqrt{\alpha b}}{1-\alpha}$, $\forall t$; at the same time, $x_{b,t} > \sqrt{\frac{b}{\alpha}}$, $\forall t$ due to (13b).*

Based on the convergence results of $\beta$-determined sequences, the following corollary allows estimating the convergence rates of $\beta$-undetermined sequences.

**Corollary 2** *Given Lemma 1, Lemma 2, and the reasoning from Remark 2, we obtain the following bounds on convergence rates of an arbitrary $\beta$-undetermined sequence* (10)*:*

1. *if $x_0 > \sqrt{\frac{a}{\alpha}}$, then $(1-2\alpha)^t \left(x_0 - \sqrt{\frac{a}{\alpha}}\right) \leq x_t - \sqrt{\frac{a}{\alpha}}$, $\forall t$;*

2. *if $x_0 > \sqrt{\frac{b}{\alpha}}$, then $x_t - \sqrt{\frac{b}{\alpha}} \leq (1-\alpha)^t \left(x_0 - \sqrt{\frac{b}{\alpha}}\right)$ while $x_t \geq \frac{\sqrt{\alpha b}}{1-\alpha}$.*

Our final important result about the $\beta$-undetermined sequences convergence is a case of convergence to the interval determined by the stationary points of the bounding $\beta$-determined sequences $x_{a,t}$ and $x_{b,t}$. We formulate it in the following proposition (see Figure 9 for an illustration).

**Proposition 5** *An arbitrary $\beta$-undetermined sequence* (10)*, given $\frac{\alpha}{1-\alpha}\sqrt{b} \leq \sqrt{a}$, converges to the following interval:*

$$\sqrt{\frac{a}{\alpha}} \leq x_t \leq \sqrt{\frac{b}{\alpha}}, \quad t \gg 1.^4$$

*Furthermore, if $x_0 > \sqrt{\frac{b}{\alpha}}$, then $x_t$ converges to the interval linearly in $\mathcal{O}(1/\alpha)$ time.*

**Proof.** Due to the first statement of Lemma 2, $x_t \geq x_{a,t} \to \sqrt{\frac{a}{\alpha}}$, hence, we deduce that the lower bound will eventually hold for $t \to \infty$. Since $\frac{\alpha}{1-\alpha}\sqrt{b} \leq \sqrt{a}$ and due to the reasoning in Remark 2, when $\sqrt{\frac{a}{\alpha}} \leq x_t$ is fulfilled, the series either stays in the stated interval (if $x_t \leq \sqrt{\frac{b}{\alpha}}$) and never crosses it or approaches it from above thanks to the upper $\beta$-deterministic bounding sequence $x_t \leq x_{b,t} \to \sqrt{\frac{b}{\alpha}}$, so the upper bound is also (asymptotically) valid.

If $x_0 > \sqrt{\frac{b}{\alpha}}$, Corollary 2 allows us to enclose $x_t$ (while it is above $\sqrt{\frac{b}{\alpha}}$) between two linear sequences converging to $\sqrt{\frac{a}{\alpha}}$ and $\sqrt{\frac{b}{\alpha}}$, respectively, with one-minus-rate proportional to $\alpha$. This is consistent with the convergence time $\mathcal{O}(1/\alpha)$ since the convergence time of linear sequences is inversely proportional to the one-minus-rate value. ∎

---

[4]These bounds are, in general, asymptotic, so, for complete correctness, $t \gg 1$ must be substituted with $t \to \infty$; however, excluding the degenerate cases, we can often observe that $x_t$ reaches the interval in finite time.

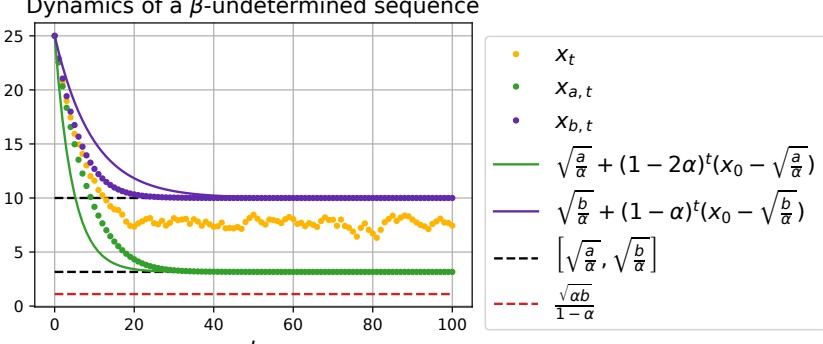

Figure 9: $\beta$-undetermined sequence (10) convergence to the $\left[\sqrt{\frac{a}{\alpha}}, \sqrt{\frac{b}{\alpha}}\right]$ interval (Proposition 5). Setting: $\alpha = 0.1$, $a = 1$, $b = 10$, $\beta_t \sim \mathcal{U}(a, b)$.

### A.5 Proof of Proposition 2

We prove Proposition 2 using the general convergence theory for so-called $\beta$-undetermined recurrent sequences of type $x_{t+1} = (1 - \alpha)x_t + \frac{\beta_t}{x_t}$, where $0 < \alpha < 0.5$ and $0 \le a \le \beta_t \le b < +\infty$, $\forall t$ (see Appendix A.4). Note that the parameters norm dynamics (3) is a special case of such a sequence with $x_t := \rho_t^2$, $\beta_t := \eta^2 \tilde{g}_t^2$, $a := \eta^2 \ell^2$, $b := \eta^2 L^2$, and $\alpha := 2\eta\lambda$ (recall that we suppress $\mathcal{O}\left((\eta\lambda)^2\right)$ terms).

**Proof.** Denote $\kappa = \sqrt{\frac{\eta}{2\lambda}}$.

In the notation of $\beta$-undetermined sequences, the condition $\rho_0^2 > \kappa\ell$ translates into $x_0 > \sqrt{\frac{a}{\alpha}}$. Thus, by applying Corollary 2, we can bound the convergence of parameters norm from below with the following linear sequence:

$$\kappa\ell + (1 - 4\eta\lambda)^t \left(\rho_0^2 - \kappa\ell\right) \le \rho_t^2.$$

The necessary $\delta$-jump condition (4a) can be equivalently reformulated as an upper bound on $\delta$:

$$\kappa\ell < \frac{\eta L}{\sqrt{2\delta}} \iff \delta < \eta\lambda\frac{L^2}{\ell^2}.$$

If this condition is fulfilled, we can estimate the minimal time required for a $\delta$-jump — the moment when the lower bound on $\rho_t^2$ intersects the $\frac{\eta L}{\sqrt{2\delta}}$ threshold. If $\rho_0^2 \le \frac{\eta L}{\sqrt{2\delta}}$, obviously, $t_{\min} = 0$, else, by solving the following equation for $t$:

$$\sqrt{\frac{\eta}{2\lambda}}\ell + (1 - 4\eta\lambda)^t \left(\rho_0^2 - \sqrt{\frac{\eta}{2\lambda}}\ell\right) = \frac{\eta L}{\sqrt{2\delta}},$$

we obtain (5).

Again, $\rho_0^2 > \kappa L$ is equivalent to $x_0 > \sqrt{\frac{b}{\alpha}}$ and, due to Corollary 2, the following upper bound on $\rho_t^2$ holds (at least while $\rho_t^2 \ge \kappa L$):

$$\rho_t^2 \le \kappa L + (1 - 2\eta\lambda)^t \left(\rho_0^2 - \kappa L\right).$$

Now, if $\delta$ is so small that the sufficient condition for a jump (4b) is fulfilled before $\rho_t^2$ converges to $\kappa L$, i.e.,

$$\kappa L < \frac{\eta\ell}{\sqrt{2\delta}} \iff \delta < \eta\lambda\frac{\ell^2}{L^2},$$

we can similarly estimate the maximal required time for a $\delta$-jump (6) as the moment when the upper bound on $\rho_t^2$ intersects the $\frac{\eta\ell}{\sqrt{2\delta}}$ threshold. ∎

## A.6   Proof of Proposition 3

As in the previous section, we prove Proposition 3 using the general theory on $\beta$-undetermined sequences (see Appendix A.4). We remarked above that the parameters norm dynamics (3) is a special case of such a sequence with parameters $a := \eta^2 \ell^2$, $b := \eta^2 L^2$, and $\alpha := 2\eta\lambda$.

**Proof.** According to Proposition 5, if for a $\beta$-undetermined sequence $x_t$ the condition $\frac{\alpha}{1-\alpha}\sqrt{b} \leq \sqrt{a}$ is fulfilled, then one can show that $x_t \in \left[\sqrt{\frac{a}{\alpha}}, \sqrt{\frac{b}{\alpha}}\right], t \gg 1$; furthermore, if $x_0 > \sqrt{\frac{b}{\alpha}}$, then $x_t$ converges to the interval linearly in $\mathcal{O}(1/\alpha)$ time. For the parameters norm dynamics, the condition $\frac{\alpha}{1-\alpha}\sqrt{b} \leq \sqrt{a}$ is equivalent (up to $\mathcal{O}\left((\eta\lambda)^2\right)$ terms) to $2\eta\lambda L \leq \ell$ as $\frac{\alpha}{1-\alpha} = \frac{2\eta\lambda}{1-2\eta\lambda} = 2\eta\lambda + \mathcal{O}\left((\eta\lambda)^2\right)$. Now, if it holds, we can apply Proposition 5 and conclude the proof. ∎

**Remark 3** *We can reformulate the same result in terms of the effective learning rate* $\tilde{\eta}_t = \eta/\rho_t^2$:

$$2\eta\lambda L \leq \ell \leq \tilde{g}_t \leq L \implies \frac{\sqrt{2\eta\lambda}}{L} \leq \tilde{\eta}_t \leq \frac{\sqrt{2\eta\lambda}}{\ell}, \ t \gg 1.$$

## A.7   Discussion on $2\eta\lambda L \leq \ell$ condition

In this section, we discuss the assumption $2\eta\lambda L \leq \ell$ made in Proposition 3, implying that the lower and the upper effective gradient norm bounds must not differ too much. First of all, we would like to remark that this condition is generally fulfilled in practice for small $\eta\lambda$ product even when the bounds $\ell$ and $L$ are taken globally, i.e., they satisfy $\ell \leq \tilde{g}_t \leq L, \forall t$. We also note that Wan et al. [26] made a very close assumption in their main Theorem 1 (Assumption 3). However, even if it is not fulfilled, our generalized parameters norm equilibrium result is still valid to some extent.

First, consider the case when $0 < \ell < 2\eta\lambda L$. Then, according to the general $\beta$-undetermined sequences theory presented in Appendix A.4, the lower bound $\kappa\ell \leq \rho_t^2$ remains valid for large $t$. If $\rho_t^2$ falls below $2\eta\lambda L$, it can potentially "hop" over the upper bound of the interval $\kappa L$. However, due to $\tilde{g}_t \leq L$ and property (13e) of $\beta$-determined sequences (see Appendix A.4.1) $\rho_t^2$ is still upper bounded by the value $(1 - \eta\lambda)^2 \kappa\ell + \frac{\eta^2 L^2}{\kappa\ell}$. Hence, globally, the parameters norm stays bounded even when $2\eta\lambda L \leq \ell$ does not hold. Furthermore, according to the second statement of Corollary 2, once $\rho_t^2$ exceeds the $\kappa L$ value, it immediately starts converging to it again. So the same $[\kappa\ell, \kappa L]$ interval of attraction is still preserved.

Now, we argue that setting $\ell = 0$, i.e., bounding the effective gradient norm from below with zero, is vacuous.[5] Again, we remark that the assumption about separating $\ell$ from zero was made, e.g., by Wan et al. [26]. Arora et al. [1] show that effective gradients (in case of learning without WD) decay sublinearly, which by itself means that in finite time horizon, it is always reasonable to set $\ell > 0$. Moreover, as we show, parameters norm evolves linearly, i.e., faster than the effective gradients; therefore, it must quickly acclimate to local $\ell$, $L$ changes and hence respect the boundaries from Proposition 3. But even based on general results on gradient-based optimization, we anticipate that, in general, $\ell$ should not approach zero. We can rewrite the expression for $\rho_t^2$ (3) in the following way:

$$\rho_t^2 = (1 - \eta\lambda)^2 \rho_{t-1}^2 + \eta^2 g_{t-1}^2 = \cdots = (1 - \eta\lambda)^{2t} \rho_0^2 + \eta^2 \sum_{t'=0}^{t-1} (1 - \eta\lambda)^{2(t-t'-1)} g_{t'}^2 = \quad (15)$$

$$= (1 - \eta\lambda)^{2t} \rho_0^2 + \eta^2 \frac{1 - (1 - \eta\lambda)^2}{1 - (1 - \eta\lambda)^{2t}} \bar{g}_t^2 \approx \{t \gg 1\} \approx (1 - \eta\lambda)^{2t} \rho_0^2 + C\bar{g}_t^2, \quad (16)$$

where $\bar{g}_t$ is an exponential moving average of the gradient norm and $C = 2\eta^3\lambda + \mathcal{O}\left((\eta\lambda)^2\right)$ is constant. It is well-known that for first-ordered methods, the lower gradient norm bound generally decays sublinearly [4]. Note that the cosine between adjacent iterates (8) depends only on the $g_t^2/\rho_t^2$ ratio. For large $t$, this ratio, due to (16), is determined only by the $g_t^2/\bar{g}_t^2$ ratio since the first term decays linearly, i.e., faster than $g_t^2$. It is reasonable to conjecture that $g_t$ oscillates around its mean value $\bar{g}_t$ hence hindering stabilization of the training dynamics which, in turn, implies that the effective gradient does not vanish. Thus, implying $\ell > 0$ seems to be a reasonable assumption.

---

[5]Excluding, perhaps, some exceptional degenerate cases when the function and hyperparameters are chosen so that the dynamics converge to a stationary point in a finite number of steps.

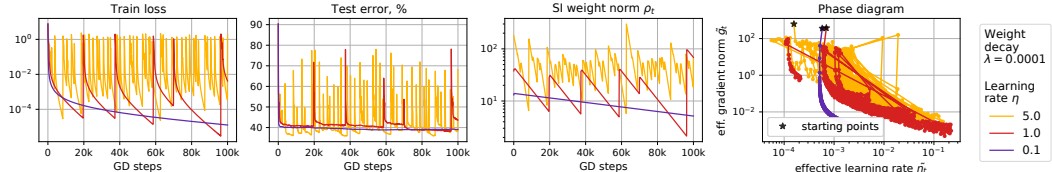

Figure 10: Periodic behavior of scale-invariant ConvNet on CIFAR-10 trained using full-batch GD with the weight decay of 0.0001 and different learning rates.

# B Experimental details

**Datasets and architectures.** We conduct experiments with two convolutional architectures, namely a three-layer convolutional neural network (ConvNet) and ResNet-18, on CIFAR-10 [15] and CIFAR-100 [16] datasets. We use the implementation of both architectures available at `https://github.com/g-benton/hessian-eff-dim`. CIFAR datasets are distributed under the MIT license, and the code is under Apache-2.0 License. To make the majority of neural network weights scale-invariant, we insert additional BN layers according to Appendix C of Li and Arora [18]. We use the standard PyTorch initialization for all layers. We use ResNet of standard width. For ConvNet, we use the width factor of 32 for fully scale-invariant networks on CIFAR-10 and the width factor of 64 for all experiments on CIFAR-100 and experiments with practical modifications on CIFAR-10.

**Fully scale-invariant setup.** Most of the experiments are conducted with the scale-invariant modifications of both architectures obtained using the approach of Li and Arora [18]. In addition to inserting extra BN layers, we fix all non-scale-invariant weights, i.e., BN parameters and the last layer's parameters. For BN layers, we use zero mean and unit variance. We fix the bias vector at random initialization and the weight matrix at rescaled random initialization for the last layer. In most of the experiments, we rescale the last layer's weight matrix so that its norm equals 10, but we discuss other scales in Appendix G.

**Training.** We train all networks using SGD with a batch size of 128 and various weight decays and learning rates. In the experiments with momentum, we use the momentum of 0.9. In the experiments with data augmentation, we use standard CIFAR augmentations: random crop (size: 32, padding: 4) and random horizontal flip. All models were trained on NVidia Tesla V100 or NVidia GeForce GTX 1080. Obtaining the results reported in the paper took approximately 1K GPU hours.

**Full-batch GD experiments.** Full-batch GD training experiments are conducted on the 4.5K-sized random subset of the train dataset. The test set in this experiment consists of 5K randomly chosen test objects.

**Logging.** In all experiments except Figures 4, 11, and 13 we log all metrics after each epoch, computing train loss and its gradients by making an additional pass through the training dataset. We log all metrics after each (S)GD step in three specified figures, computing train loss and its gradients over a batch.

# C Full-batch gradient descent

In the main paper, we presented the periodic behavior results for SGD. In this section, we show that the periodic behavior is observed for full-batch GD training and hence is not a consequence of stochastic training. We replicate all experiments of Section 4: Figure 10 visualizes training dynamics for different learning rate values, Figure 11 presents a closer look at one period of training (see also Figure 13 for the plots of cosines between adjacent steps), and Figure 12 replicates the ablation experiment with fixing the weight norm. All the effects discussed in the main text for the SGD case hold for the GD case. We note that phase $B$ is longer for full-batch GD training because the absence of stochasticity allows stable training at lower train loss, and destabilization occurs later.

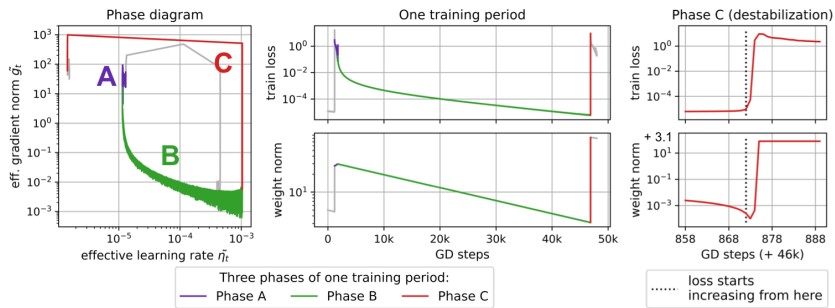

Figure 11: A closer look at one training period for scale-invariant ConvNet on CIFAR-10 trained using full-batch GD with weight decay of 0.001 and the learning rate of 0.5. Three phases of the training period are highlighted.

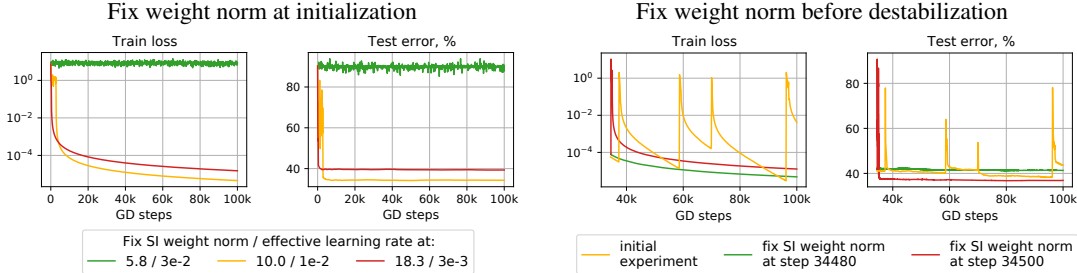

Figure 12: The absence of the periodic behavior for training with the fixed weight norm. Scale-invariant ConvNet on CIFAR-10 trained using full-batch GD with weight decay of 0.0001 and learning rate of 1.0. Left pair: the weight norm is fixed at random initialization of different scales. Right pair: the weight norm is fixed at some step of regular training before destabilization.

## D   Bounds on the effective gradient norm and $\delta$-jumps

In Section 4, we compared cosine distance between weights at adjacent SGD steps of phase $B$ with theoretically derived bounds for $\delta$-jumps from Section 5.1. In Figure 13, right pair, we present a similar comparison for the full-batch GD case: the effect of both bounds and the cosine metric itself growing in the second half of the phase is even more prominent for the GD case than for SGD. Below we describe how we choose the local bounds $\ell$ and $L$ on the effective gradient norm $\tilde{g}_t$ which are used in the theoretical bounds. All bounds are visualized in Figure 13.

In both GD and SGD cases, we chose $\ell(t)$ and $L(t)$ as smooth functions of $t$. Note that taking such dynamical bounds does not contradict our theoretical results (see Remark 1). For the SGD case, we chose $\ell(t) = \frac{c}{t-t_0}$ and $L(t) = \frac{C}{t-t_0}$, where $t_0$ is the first iteration of the considered training period. For the GD case we used the same approach, but had to take $\ell(t) = \frac{c}{(t-t_0)^2}$ to better mimic the behavior of the lower envelope of the effective gradients norm. We handpick constants $0 < c < C$ and iteration $t_{\text{valid}}$ separately for SGD and GD cases so that

$$\ell(t) \leq \tilde{g}_t \leq L(t) \tag{17}$$

for all $t \geqslant t_{\text{valid}}$ in phase $B$.

## E   Optimization of common scale-invariant functions with weight decay

In this section, we show that periodic behavior may be observed not only when training neural networks but also during gradient decent optimization of common scale-invariant functions with weight decay and a constant learning rate. As an example we consider a function of two variables $f(x, y) = \frac{x^2}{x^2+y^2}$, which is naturally scale-invariant. The minimum value of $f$ equals 0 and is achieved at any point with $x = 0$.

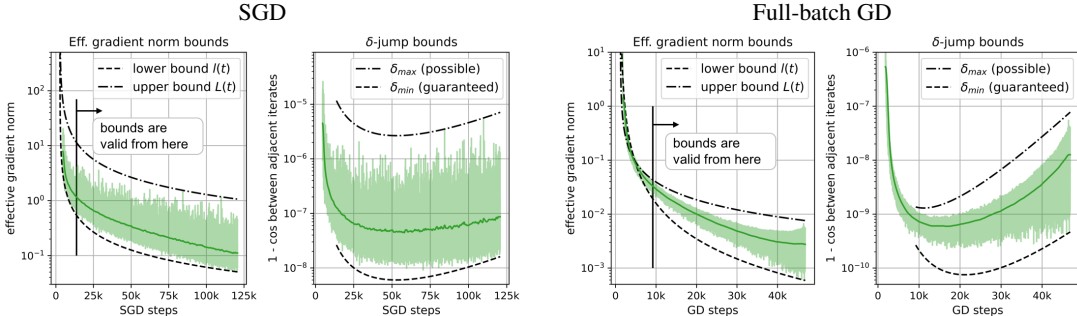

Figure 13: Effective gradient norm and cosine distance between weights at adjacent (S)GD steps, presented along with their smoothed trends. Phase $B$ of one period of training scale-invariant ConvNet on CIFAR-10 is shown. Weight decay / learning rate: 0.001 / 0.01 for SGD, 0.0001 / 0.5 for GD. $\delta$-jump bounds are obtained using the bounds on the effective gradient norm.

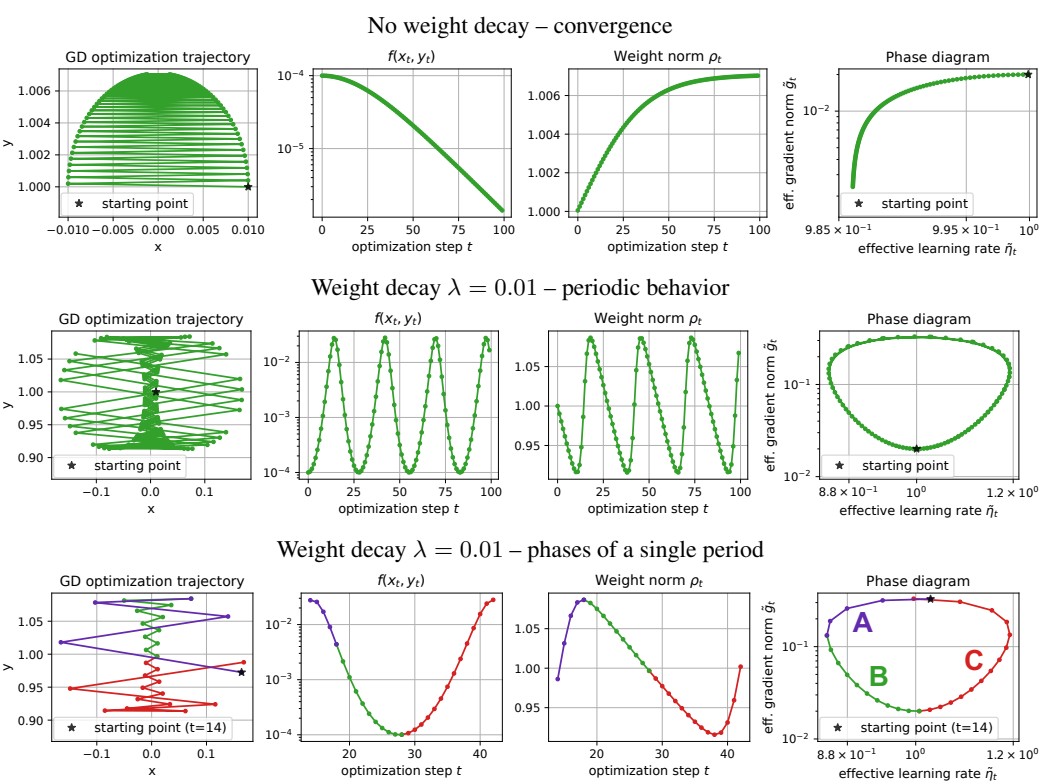

Figure 14: Minimization of a simple scale-invariant function $f(x, y) = x^2/(x^2 + y^2)$ with and without weight decay. For all experiments the initial point $(x_0, y_0) = (0.01, 1.0)$, learning rate $\eta = 1$.

If we minimize $f$ without weight decay, the optimization procedure converges to a stationary point since its effective learning rate monotonically decays, as can be seen in the top row of Figure 14. This behavior accords with the results of Arora et al. [1].

However, with weight decay we can observe the same periodicity of the optimization dynamics as demonstrated by experiments with neural networks (see the middle row of Figure 14). Moreover, in this case, the optimization experiences the same three phases in the period (see the bottom row of Figure 14, which is analogous to Figure 4 in the main text).

This confirms that the periodicity of optimization dynamics is a general property of scale-invariant functions optimized with weight decay and is not specific to neural networks.

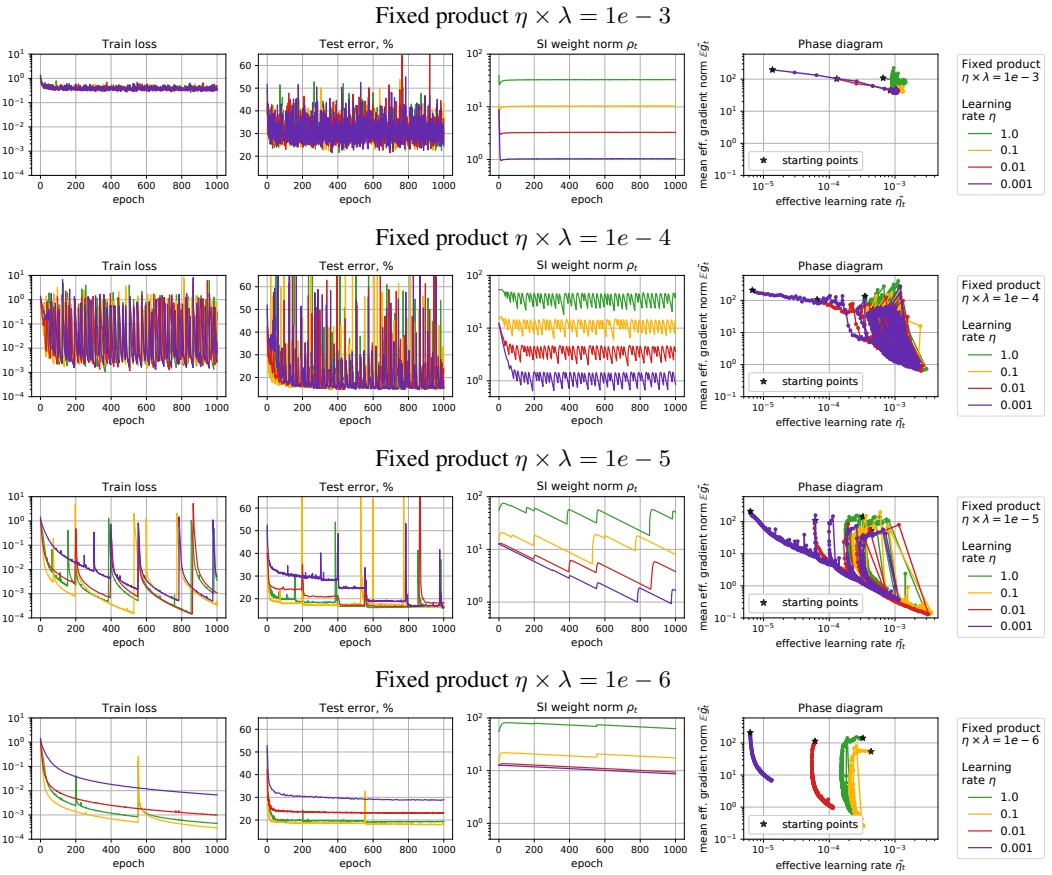

Figure 15: Training dynamics of scale-invariant ConvNet on CIFAR-10 trained with fixed learning rate – weight decay products. Axes limits are the same in each column for convenient comparison.

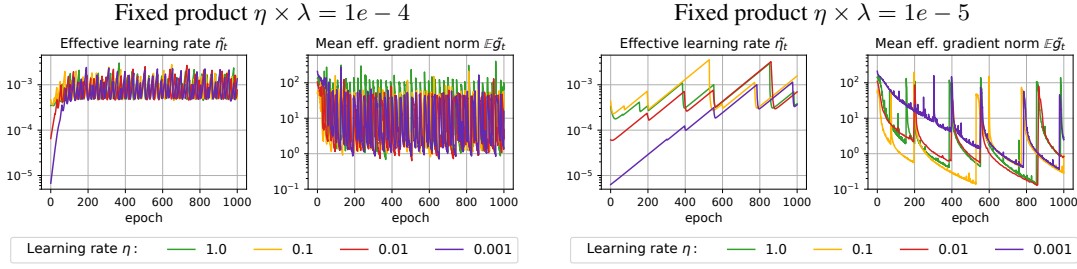

Figure 16: A closer look at dynamics of the effective learning rate and mean effective gradient norm of scale-invariant ConvNet on CIFAR-10 trained with two different fixed learning rate – weight decay products. Axes limits are the same for corresponding metrics for convenient comparison.

# F Influence of learning rate and weight decay on the periodic behavior of scale-invariant networks

## F.1 Fixed learning rate – weight decay product

In this section, we discuss the effect of the learning rate – weight decay product on the training process. Figure 15 visualizes training progress for different values of the product (plot rows) and variable ratio of two specified hyperparameters (different lines in each row). We observe that training converges to similar consistent behavior with the fixed learning rate – weight decay product. Specifically, the frequency of the periods, the minimal achieved train loss and test error, and the ranges of the effective

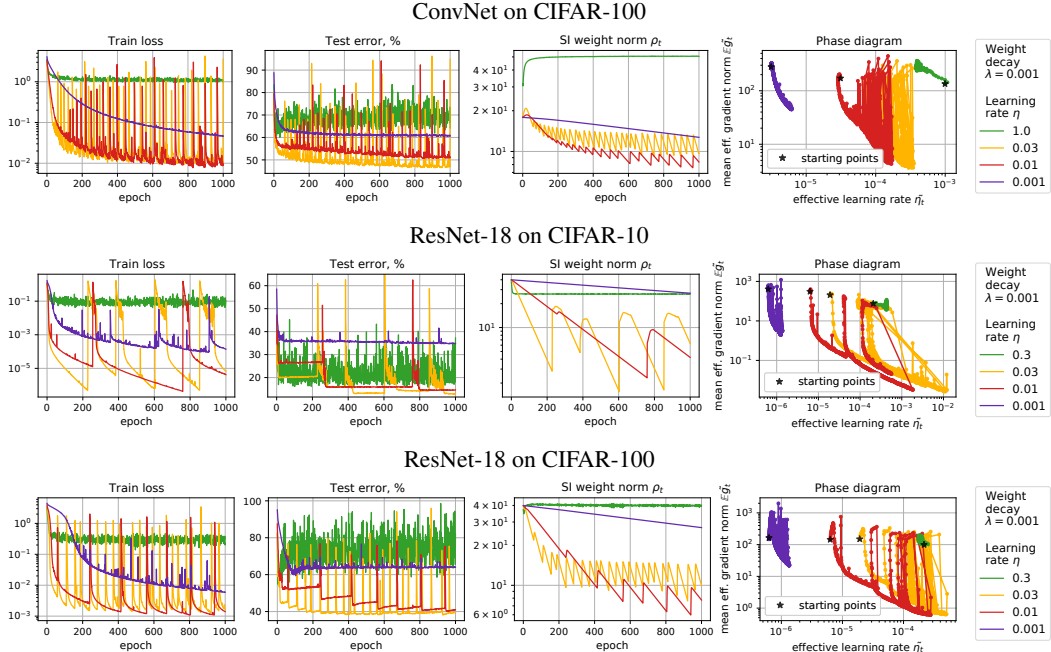

Figure 17: Training dynamics of scale-invariant networks trained with fixed weight decay and different learning rates.

gradient norm and the effective learning rate are similar across different lines in one row. The last-mentioned ranges are visualized in more detail for selected setups in Figure 16. The described empirical results agree with Remark 3 in Appendix A.6. Mainly, the remark states that with a fixed learning rate – weight decay product and bounded effective gradient norm, training converges to a bounded effective learning rate, and the effective learning rate bounds depend only on the effective gradient norm bounds. In practice, we observe that the last-mentioned bounds are similar across different ratios of weight decay and learning rate (see Figure 16). Thus, the effective learning rate bounds are also similar across different ratios (see Figure 16).

However, although the characteristics of the *consistent* periodic behavior are similar across different ratios of the learning rate and the weight decay when their product is fixed, the length of the *warm-up* stage may vary. The reason is that we use the standard initialization for all networks, i.e., the same initial weight norm for all combinations of hyperparameters. At the same time, given different ratios of weight decay and learning rate, the weight norm converges to different ranges (see Figure 15 and Proposition 3). The final weight norm may substantially differ from the initial weight norm, and the larger the difference, the longer the warm-up stage.

We note, however, that, according to Proposition 4 in Appendix A.1, if we fixed the direction of initialization (i.e., the point on the unit sphere) and then appropriately rescaled it (proportionally to the square root of the learning rate), the training dynamics would be exactly the same for different ratios of learning rate and weight decay, given their product is unchanged, including the warm-up stage.

## F.2 Fixed weight decay and different learning rates

Figure 17 supplements Figure 3 and shows how the learning rate affects the periodic behavior for different dataset-architecture pairs when the weight decay is fixed. For CIFAR-100, we had to increase the ConvNet's width factor up to 64 and the last layer's weight norm up to 20 to ensure the network is able to learn the train dataset and achieve low train loss. The general picture is the same as described in Section 6: the periodic behavior is absent for too low or too high learning rates and present for a range of learning rate values, which also allow lower test error. Interestingly, for ResNet on CIFAR-10 with the learning rate of 0.03, phase $A$ is noisy and quite long because of the relatively

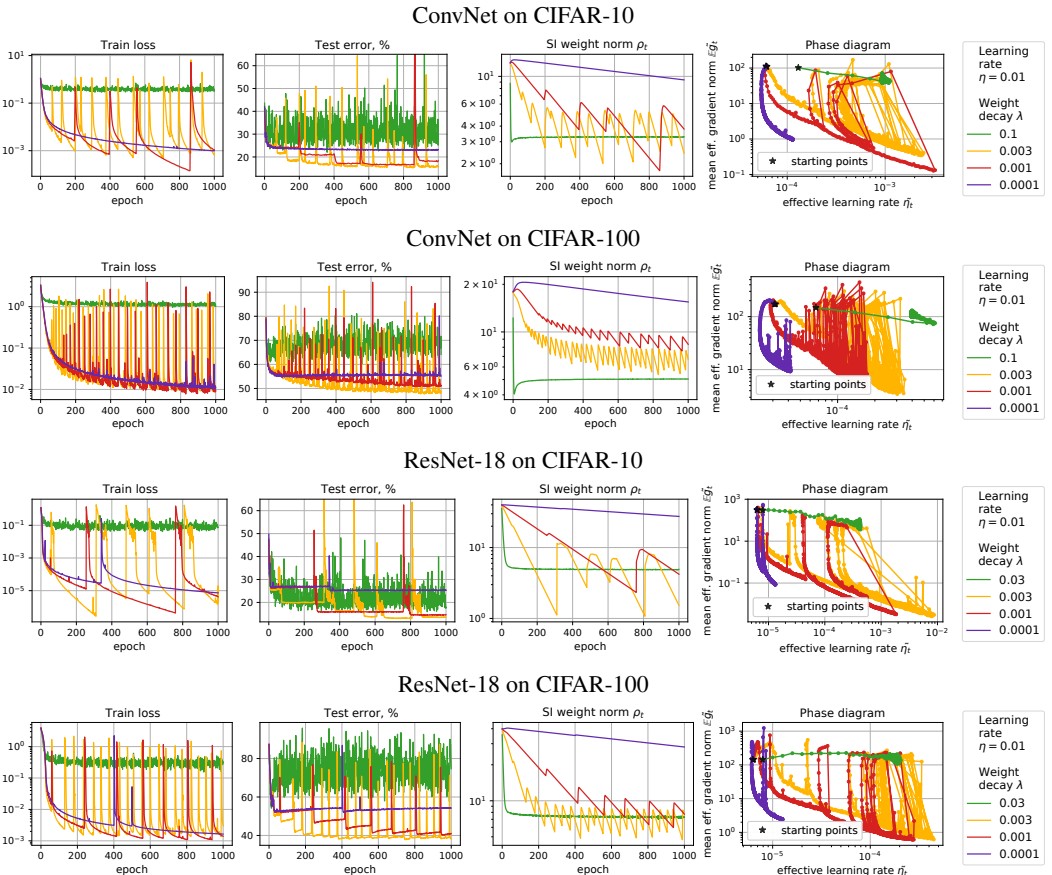

Figure 18: Training dynamics of scale-invariant networks trained with the fixed learning rate and different weight decays.

high learning rate, but training still proceeds to phase $B$, while for larger learning rate, training gets stuck at high train loss.

### F.3 Fixed learning rate and different weight decays

Figure 18 shows the periodic behavior when the learning rate is fixed, and the weight decay is varied for different dataset-architecture pairs. The general observations are the same as when the learning rate is varied with the fixed weight decay. Notably, the periodic behavior is absent for too low or too high weight decay coefficients and present for a range of weight decay values, which also allow reaching lower test error. Further, using a larger weight decay increases the frequency of the periods.

## G    Influence of the last layer weight matrix norm

In scale-invariant neural networks, we fix the weights of the last layer. Moreover, we renormalize the weight matrix to the specified weight norm, which becomes a new hyperparameter. This hyperparameter determines the level of the neural network's confidence in its predictions, and, in the main text, we set it to a large value (10) to achieve high confidence and to make our setup closer to the conventional neural network training (when all parameters are trained). In this section, we discuss the influence of the specified hyperparameter on periodic behavior.

Figure 19 shows results for ConvNet on CIFAR-10 and ResNet on CIFAR-100 and different values of the last layer's weight norm. The lowest presented last layer's weight norms are close to the norms obtained at random initialization without rescaling. Using low last layer's weight norm leads to low network's confidence which prohibits reaching low train loss and may result in the absence of

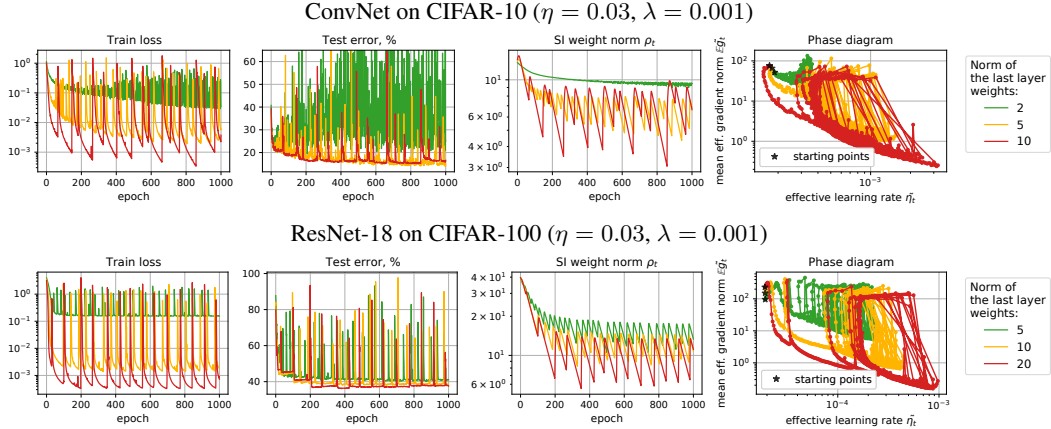

Figure 19: Influence of the last layer weight matrix norm on the periodic behavior.

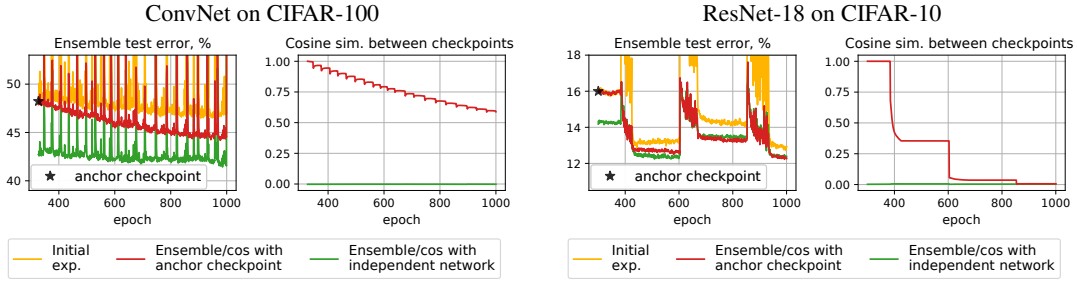

Figure 20: Similarity in the weight space (cosine sim.) and in the functional space (ensemble test error) for different checkpoints of training scale-invariant ConvNet on CIFAR-100 (left pair) and ResNet on CIFAR-10 (right pair) using SGD with weight decay of 0.001 and learning rate of 0.03.

the periodic behavior. In the main text, we use larger values of the last layer's weight norm, which circumvents this issue.

## H  Minima achieved at different training periods

Figure 20 supplements Figure 6 for analyzing the weight/functional similarity of optima achieved at different training periods. The general observations are the same as in Section 6. Interestingly, the ensemble of two models spawned by optima from different periods can reach the error of two independent networks ensemble for both architectures on the CIFAR-10 dataset and does not reach one on the CIFAR-100 dataset (in given epochs budget).

## I  Practical modifications

Figure 21 supplements Figure 7 and shows the presence of the periodic behavior in a more practical setting, i.e., with trainable non-scale-invariant parameters, momentum, and data augmentation, for ConvNet on CIFAR-100 and ResNet on CIFAR-10 and CIFAR-100. For a more detailed discussion, see Section 7 in the main text.

We also consider training neural networks with a more sophisticated optimizer, Adam [13], and show the presence of the periodic behavior for ConvNet on CIFAR-10 in Figure 23.

In order to show that our results extrapolate to other normalization approaches besides batch normalization, we train ConvNet on CIFAR-10 using layer normalization [2] and instance normalization [24] and demonstrate the presence of the periodic behavior in this setting in Figure 22.

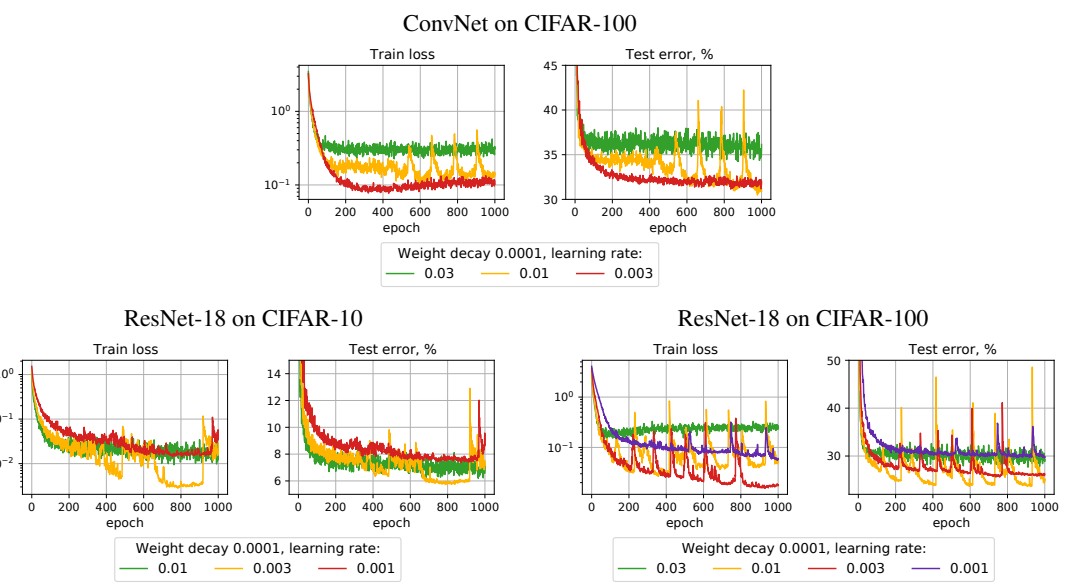

Figure 21: Training dynamics of networks trained with more practical modifications, i.e., with learnable non-scale-invariant parameters, momentum, and augmentation (all modifications together).

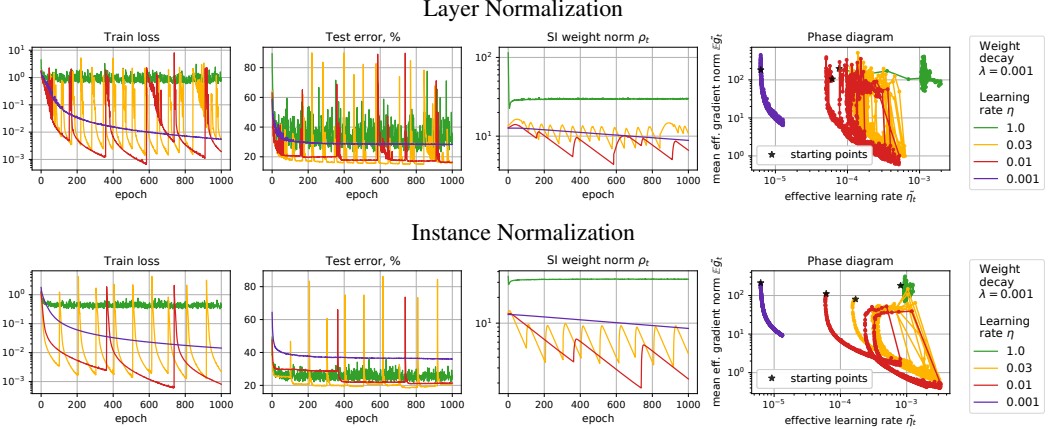

Figure 22: Training dynamics of scale-invariant ConvNet with other normalization approaches on CIFAR-10.

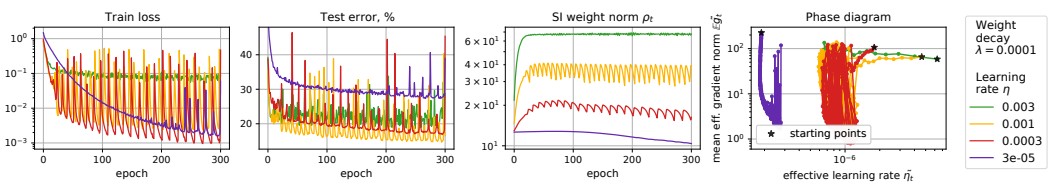

Figure 23: Training dynamics of scale-invariant ConvNet on CIFAR-10 trained using Adam.

# J  Comparison with previous works

In this section, we compare our experimental setup with that of the prior art and point out the main factors for why previous experiments mostly do not show periodic behavior.

As we stated in Section 7, periodic behavior is not usually observed when training normalized neural networks due to a relatively small epochs budget and usage of learning rate schedules. Moreover, some hyperparameters settings can make periods too slow or even unreachable, which both hinder observation of the periodic behavior in practice (see, e.g., the smallest and the highest learning rate curves in Figure 3). Finally, the use of data augmentation and/or models that are too simple to learn a given dataset does not allow even the first period to be completed within a reasonable time frame. These are the key reasons periodic behavior was mainly not reported in the literature previously. Below we discuss the particular aspects of several most related works.

One of the works closest to ours, Li et al. [19], discovers the unstable behavior of full-batch GD training of scale-invariant networks and at the same time reports convergence to a constant equilibrium when training with SGD. We suppose that the experiments of Li et al. [19] with full-batch GD depict exactly our periodic behavior. Speaking of SGD experiments, we suspect that, despite a large epochs budget, Li et al. [19] did not encounter periodic behavior in most of their experiments due to data augmentation, different hyperparameters settings, and learning rate schedules. In other words, they mainly observed a prolonged phase $A$ in their experiments without reaching the end of even the first period, which may seem like convergence to a stable equilibrium.

Wan et al. [26], who also study the convergence of scale-invariant parameters dynamics to the equilibrium (which, however, is now *dynamical*, i.e., depends on the behavior of effective gradients, in par with our work), did not find periods in their experiments as well. This can also be attributed to data augmentation and learning rate schedules but most importantly to short training, which does not allow finishing phase $A$ of the first period, as seen by the increasing effective gradients norm throughout training in Figure 2 therein.

As mentioned in the main text, Li et al. [17] discovered that training weight-normalized neural networks with improperly selected weight decay may become unstable and even result in training failure since the numerical gradient updates are beyond the representation of float. This is the extreme case of destabilization in phase $C$ when scale-invariant parameters approach the origin too close and the gradients blow up so that training is already unable to recover due to numerical issues. In our experiments, such situations did not occur, however, we hypothesize that they can be encountered when training very large networks equipped with both weight normalization and feature normalization, which may amplify the destabilization effect of approaching the origin. Other experiments of Li et al. [17] did not reveal the periodic behavior for the same reasons as above: data augmentation, insufficient training duration, and learning rate schedules.