# OpenReview forum: "On the Periodic Behavior of Neural Network Training with Batch Normalization and Weight Decay"
_NeurIPS.cc/2021/Conference — NeurIPS 2021 Poster_

### Official Review · Reviewer_JkC9 · 2021-07-15

**Rating:** 7
**Confidence:** 4

**Summary:**

This paper studies an interesting contradiction about neural networks with Batch Normalization (BN) and weight decay (WD): some works show that BN and WD can cause instabilities in training, while other works show convergence to an equilibrium that stabilizes training metrics. After conducting very extensive experiments, the authors claim that the training process converges to a consistent periodic behavior, which regularly exhibits instabilities and restarts a new period of training.

**Limitations And Societal Impact:**

As stated in Section 6, the main limitation of this work is that periodic behavior mainly occurs on fully scale-invariant neural nets training for a sufficiently long time. Nevertheless, the training dynamics when $t \to \infty$ is a fundamental problem to study in the theory of deep learning, and the authors made a good contribution to this problem in the setting they adopted.

**Main Review:**

I really appreciate the authors' work for their extensive experiments across different settings. The experiment setups are stated clearly and the figures are also very clear. I learned a lot from those experiments and I'm convinced that the periodic behavior could be more general than the instabilizing view and the equilibrium view from previous works. I would like to vote for acceptance.

For the theory part, the propositions in the current form are not very satisfactory. Note that the theory is expected to explain the periodic behavior, rather than just the first period. This means $\rho_0$ in Proposition 2 should be interpreted as the initial norm at the beginning of each period and thus cannot be set arbitrarily except for the first period. Thus, a better theorem should upper and lower bounds $\rho_0$ by analyzing the dynamics of previous periods and provide bounds for the time period without $\rho_0$.

Regarding why this work and Li et al. [17] produce different results, I suspect the most important factor could be the noise. When the noise is large and the learning rate is small, the SDE approximation in Li et al. [17] is valid and the convergence to equilibrium should occur. However, this paper turns off the data augmentation in most of the experiments, and thus the noise may not be sufficient.

I believe it could greatly benefit future research if the authors can compare their experiment setups with previous works. Section 6 has already done a very good job, but it could be more helpful if the authors can point out the most important factor for why previous experiments do not show this periodic behavior, specifically for each the works they are directly following (e.g., Li et al. [15], Li et al. [17]).

(The authors sometimes omit many second-order terms like  $O(\delta^2)$ or $O((\eta \lambda)^2)$ in the derivation. I would suggest the authors use $\approx$ and $\lesssim$  instead of $=$ and $\le$ when such approximation is used. E.g., Proposition 1.)

-----------------

The rebuttal addressed most of my concerns. Although the theory part in the current form is not very strong, the empirical study in this paper is very nice and the period behavior is very interesting.  I would like to keep my score for acceptance and I look forward to seeing the camera-ready version of the theory part.

**Time Spent Reviewing:**

4

---

> ### Author Response · Authors · 2021-08-10
> **Authors response to Reviewer JkC9**
>
> Thank you for the valuable feedback! We are very glad that you find our work interesting. Please allow us to address some of your concerns.
>
> * “the propositions in the current form are not very satisfactory”
> We would like to elucidate that our propositions are not stated in terms of obligatory constraints on $\rho_0$ (i.e., “in order to achieve these properties, set the initial norm to this value”) but rather as certain guarantees provided if $\rho_0$ meets some conditions (e.g., “$\delta$-jump is guaranteed to occur no sooner than X once weight norm satisfies Y”). To put it simply, you may take the beginning of some period (not necessarily the first one) and measure the weight norm. Then you can anticipate specific effects stated in our propositions if it satisfies certain conditions. If it does not, then you could relax the constraints, e.g., loose bounds on the effective gradient norm, and obtain a weaker result or wait until the norm decays and the conditions are met and take that point as $\rho_0$. We will clarify this moment in our text and probably reformulate the statements to avoid possible misinterpretation.
> * Comparison with Li et al. [17]
> We suspect that Li et al. [17] did not observe periodic behavior (PB) in most of their experiments due to data augmentation (DA), different hyperparameters settings, and learning rate schedules, which all hinder the observation of PB in practice. Therefore, we agree that DA is partially responsible here, however, not due to a closer connection with the underlying SDE (which might not be true, see, e.g., https://arxiv.org/abs/2102.12470, where the authors deliberately turned off DA and Batch Normalization in some of their experiments to achieve a more accurate approximation of the SDE flow) but due to increasing complexity of the dataset which automatically prolongs phases A and B. Interestingly, PB is clearly visible in the experiment with GD of Li et al. [17] (see Figure 1 in https://arxiv.org/abs/2010.02916). Also, note that the theoretical results of Li et al. [17] on reaching the equilibrium are based on a very strong assumption that the effective gradient variance becomes approximately constant (Lemma 5.2) which, as we found in our experiments, does not generally hold.
> * “I believe it could greatly benefit future research if the authors can compare their experiment setups with previous works”
> This is a very good suggestion, thank you! We will add such a comparison in Appendix.
> * On notation
> We agree with you that it is formally incorrect to write strict signs like $=$ when omitting certain terms. We will fix our notation.
> * “periodic behavior mainly occurs on fully scale-invariant neural nets training for a sufficiently long time”
> As we have mentioned in our text and other replies, PB can be clearly observed in networks with non scale-invariant parameters as well (see, e.g., Figure 1), however, to prove our point that this effect must be attributed mainly to the interplay between the normalized parameters and weight decay, we mostly study it in the distilled setting when the whole network is scale-invariant.

---

### Official Review · Reviewer_ERRM · 2021-07-16

**Rating:** 7
**Confidence:** 4

**Summary:**

The paper shows that commonly used Batch Norm and Weight Decay on networks with frozen last layer exhibit periodic behavior in training. The authors provide theoretical guarantees and also study extensions of their results to more commonly used settings and show where their results break down.

**Limitations And Societal Impact:**

Yes

**Main Review:**

Building upon some previous work in the theory of deep learning field, the authors show both theoretically and experimentally that for a network trained without last layer, Batch Norm and Weight Decay have opposing actions on control of the norm, leading to a periodic behavior in training.

It is interesting that despite this rather atypical setting, the authors then expand their experiments and relax the assumptions one by one to reach the more typical regime of training. It is interesting to note that the periodicity of the behavior is broken by momentum, and not visible unless one uses a relatively large learning rate, which is something this reviewer has previously also noticed while training NNs with large learning rates on many occasions.

It would be interesting to see how these results coexist with the setting of all layers being trained, where it is expected that Neural Collapse (see V. Papyan et al. "Prevalence of neural collapse during the terminal phase of deep learning training"), and hence convergence, happens when we train deep nets with both Batch Norm and Weight Decay, see for example:
- A. Rangamani, "Dynamics and Neural Collapse in Deep Classifiers trained with the Square Loss"
- T. Ergen and M. Pilanci,  "Revealing the structure of deep neural networks via convex duality"


**Time Spent Reviewing:**

5

---

> ### Author Response · Authors · 2021-08-10
> **Authors response to Reviewer ERRM**
>
> Thank you for the valuable feedback! Please allow us to address some of your concerns.
>
> * “periodicity of the behavior is broken by momentum”
> In fact, momentum does not break the periodic behavior and even speeds up the periods, see Figure 6 (left plot, yellow line). Additionally, periodic behavior is also present in the experiments with Adam, which we conducted after the Reviewer 6wdK’s question.
> * “It would be interesting to see how these results coexist with the setting of all layers being trained”
> We do consider such a setting in our experiments (e.g., all experiments from Figure 1 and green line from Figure 6, left). We can observe that training the last layer does not change the periodic behavior itself but rather influences its frequency (apparently due to different scaling of softmax temperature).
> * On Neural Collapse
> We have indeed studied this effect in our preliminary experiments. In the experiments with trainable last layer weights, we detected that Neural Collapse (NC) in terms of almost complete stabilization of the last layer emerges shortly after the end of phase A of each cycle, i.e., when the training error first reaches its minimum, in accordance with the findings of the initial manuscript on NC (https://arxiv.org/abs/2008.08186). In phase C, however, the last layer experiences destabilization along with other network weights. Note that in works on NC training is usually stopped at the beginning of phase B while we train our models significantly longer.

---

### Official Review · Reviewer_VdoN · 2021-07-17

**Rating:** 5
**Confidence:** 4

**Summary:**

In this paper, the authors look at scale-invariant networks trained with SGD using both batch normalization and weight decay. They study the contradiction between the equilibrium presumption and the instability presumption and claim that the training process converges to a consistent periodic behavior which includes some instabilities during training but does not lead to failure. They define the notion of delta-jumps for the dynamics of training and conjecture that the delta jumps are necessary for destabilization of the dynamics. Extensive empirical experiments are provided for supporting the claims.


**Limitations And Societal Impact:**

The authors have discussed the limitations of the work and the is no potential negative societal impact.

**Main Review:**

The paper is written clearly in most parts and the periodic observation is interesting. I also appreciate the extensive experiments. My main concern about this paper is that the observation is limited to a very specific setting and this periodic behavior is usually not observed in practice. Therefore, I believe the problem is not well motivated in the first place and even for this special case, the theory to support the claims is not complete. I am not convinced by the authors’ conjecture that delta-jumps are necessary for training dynamics destabilization. I also have the following comments/questions:

1. It is noted in [1] that destabilization can happen due to the increase in sharpness of the network (the maximum eigenvalue of the training loss Hessian). Have you done any observations regarding this?

2.  Lines 108 to 121: the explanation is insightful but very intuitive and without any figures or any exact expressions. It would be great if you could write the exact expressions and relations. For example, what do you exactly mean by effective learning rate? \frac{\eta}{\norm{x_t}^2} \proportional \sqrt{eta lambda}? or Maybe you could show the centrifugal and centripetal forces on a plot.

3. It would be great if you could discuss how limiting the assumption of bounded effective gradients is since this is the key element in generalizing proposition 3.

4. There is no Figure 7.  Please fix the number.



[1] https://arxiv.org/pdf/2103.00065.pdf


**Time Spent Reviewing:**

5

---

> ### Author Response · Authors · 2021-08-10
> **Authors response to Reviewer VdoN**
>
> Thank you for the valuable feedback! Please allow us to address some of your concerns.
>
> First, we would like to comment on some general points from the review:
>
> * “My main concern about this paper is that the observation is limited to a very specific setting and this periodic behavior is usually not observed in practice. Therefore, I believe the problem is not well motivated”
> This research was started as a study of the properties of scale-invariant nets in a long training perspective. The periodic behavior was a surprise for us and we did not find an explanation for it in the literature. Hence, we hope our work will help other researchers who study the properties of the neural networks training dynamics to better understand the observed behavior. While we conducted most of our experiments in a particular fully scale-invariant setting (note that it was also employed in a line of previous works) to investigate exactly the effect of the interplay between scale invariance and weight decay, we specifically show in Section 6 that it can actually be observed in more practical scenarios with momentum, data augmentation, and neural networks incorporating non scale-invariant weights if trained long enough. Thus, in addition to examining the properties of scale-invariant nets, which have received a lot of attention in recent studies, we believe that our results could be relevant in practical scenarios as well.
> * “I am not convinced by the authors’ conjecture that delta-jumps are necessary for training dynamics destabilization”
> While we cannot formally prove this statement (at least in the general setting of arbitrary scale-invariant neural networks), we still consider it a reasonable assumption. Effectively, we say that a “well-behaved” function cannot change significantly until we somewhat notably change its argument. By “well-behaved” we can assume, for instance, that its Lipschitz constant is bounded (at least locally), which makes sense for networks with Batch Normalization (https://arxiv.org/abs/1805.11604). Note, however, that even if the loss of scale-invariant network has “unstable” regions on a unit hypersphere with very high or even unbounded Lipschitz constant, our analysis is still relevant. Making larger steps in such regions would lead to a higher chance of divergence. The main point of our analysis is that the closer we approach the origin, the larger effective steps (steps on a unit hypersphere) we make, thereby paving the way for destabilization.
>
> Answers to your questions:
>
> 1. In [1] authors raise very interesting questions. The increase in sharpness of the network indeed indicates instability in the training but why this progressive sharpening occurs is still an open question. In our paper, we investigate the reasons for the training destabilization in a particular case of networks with BN and WD. In this case, the interplay between BN and WD leads training to the unstable zone near the origin where the sharpness of the network increases. In our experiments, we have measured several additional metrics responsible for the model stability, like the trace of the Fisher Information Matrix, which is very close to Hessian but easier to compute (https://arxiv.org/abs/1906.07774). We found that all of them behave similarly to the norm of the effective gradient that we plot in our phase diagrams, namely, they grow during destabilizing phase C and then slowly decrease during phases A and B. We argue that in the case of networks with BN and WD the main reason for the increase in these metrics is a decrease in weight norm. Based on the results of [1], there may be other reasons for unstable training behavior and other unstable zones for different neural networks. Thus, further studies are needed. We are inspired by this research direction and hope that our study makes a step towards understanding such an intriguing behavior.
> 2. We will try to make the text and the plots more clear. Basically, the given intuition is based on concrete properties of scale-invariant functions (2a and 2b) that make their gradients orthogonal to the radial direction and scale inversely proportional to the parameters norm, thus providing the “centrifugal force”, and the natural “centripetal force” of the weight decay. The concrete definition of the effective learning rate is provided in line 200 $\tilde{\eta}_t = \eta / \rho_t^2$, it effectively incorporates the second property. We will put it before lines 108-121.
> 3. Since the norm of the effective gradient is always bounded from below by zero and loss functions of networks with Batch Normalization are typically smooth (https://arxiv.org/abs/1805.11604), it seems reasonable to assume that there exist some values $\ell$ and $L$ such that $0 \le \ell \le \tilde{g}_t \le L < +\infty$. The main assumption in Proposition 3 is that these bounds are somewhat close, i.e., $2 \eta \lambda L \le \ell$. We discuss this assumption in detail in Appendix A.7. Briefly, this condition is not burdensome and almost always fulfilled in practice even if $\ell$ and $L$ are set as global bounds on the effective gradient norm for the whole training process (note that generally, for our results to be as accurate as possible, $\ell$ and $L$ are expected to be local bounds valid during the current period or in the local loss landscape region).
> 4. Thank you for noting, we will correct this typo.

---

### Official Review · Reviewer_6wdK · 2021-07-22

**Rating:** 7
**Confidence:** 3

**Summary:**

The paper tackles the issue of explaining periodic behavior that can occur when jointly training neural networks using batch normalization and weight decay, and that causes training to alternate between phases of instability and stability.
The authors give a theoretically reasonable explanation of why this phenomenon occurs. They derive the notion of delta-jumps in parameter space that are responsible for training instabilities, and derive theoretical bounds based on parameter norm, effective gradient norm, and learning rate.
They investigate this mechanism empirically by training scale-free CNNs on the CIFAR-10 and CIFAR-100 dataset classification task, and also extend their analysis to more practical scenarios, showing that the periodic behavior can still be produced in less artificial settings.

**Limitations And Societal Impact:**

Yes

**Main Review:**

The paper seems to provide a significant contribution in that it resolves the standing contradicting between the “equilibrium point vs. instability” view of training with batch normalization and weight decay. However, while the authors themselves point out that this effect can in principle also be observed in non-scale free networks, they make clear that it is of limited practical relevance due to the narrow parameter range in which it can occur.
The paper outlines its general ideas in a comprehensible way. However, some of the connections between theory and experiment could be made clearer, e.g. how the theoretically derived size of delta-jumps, convergence times etc. relates to the outcomes of their experiments. The rightmost plot in figure 3 was also a little confusing. The results are also generally shown without any statistics.

Some questions that could be clarified:

1. What happens for optimizers that use adaptive learning rates, such as Adam?
2. With what consistency does the loss achieve a lower minima after a phase of instability? Figure 6 makes it seems like this is the case, while in Figure 4 the result looks different.
3. “Making an SGD step in the direction of the loss gradient always increases the norm of scale-invariant parameters, while WD aims at decreasing the weight norm.” It is not directly apparent what this is based on. However, as this is the theoretical basis for the entire paper, it would be good to make the citation clearer and explain the mathematical reasoning in more detail.

Some minor comments:
- Using either epoch or SGD step (or making clearer how they relate to each other) consistently in the x-axis of figure 2+3
- Warm-up phases can't be seen too well from the plots
- coarse grain choices of learning rate. What happens in the range between 0.01 and 0.001?
- With which probability do the delta jumps that lead to instabilities actually occur? Do they happen with certainty if curve is in given interval, and does it always happen?
- The description of the ensemble method in 5) was a little confusing

**Time Spent Reviewing:**

12 hours

---

> ### Author Response · Authors · 2021-08-10
> **Authors response to Reviewer 6wdK**
>
> Thank you for the valuable feedback! Please allow us to address some of your concerns.
>
> First, we would like to comment on some general points from the review:
>
> * "it is of limited practical relevance due to the narrow parameter range in which it can occur"
> Although the periodic behavior (PB) could be clearly observed for a limited range of hyperparameters in the conventional setting, we would like to point out that the model generally achieves its best performance exactly in this range, so the effect is relevant to the practical setting, in our opinion.
> * “some of the connections between theory and experiment could be made clearer”
> We will provide more discussion on the connection between our theoretical and experimental results in the text. Here we briefly outline the main points. Our theoretical results are based on a conjecture that the model has to start making substantial steps on the unit sphere — $\delta$-jumps — in order to diverge. We derive several propositions regarding the properties of these $\delta$-jumps, e.g., Proposition 1: the necessary and sufficient conditions on the parameters norm for the $\delta$-jump to occur assuming that the effective gradient norm is between some values $\ell$ and $L$. For the current value of the parameters norm, this result could be equivalently reformulated in terms of minimal and maximal $\delta$-jump possible, which we plot in the rightmost panel of Figure 3. We demonstrate that the actual cosine distance between adjacent iterates indeed stays between the theoretically derived bounds. Also, from Proposition 2 we derive Corollary 1 that explains why periods become more frequent when $\eta \times \lambda$ product increases.
> * “The results are also generally shown without any statistics”
> We conducted experiments with different random seeds and the training behavior was consistent: periods frequency, minimal loss value, range of weight norms to which the training converges, and length of the warm-up period were similar for different runs. We did not include plots for all runs due to the page limit. We believe that multiple single-run results obtained with various hyperparameters values also provide solid empirical support for our claims.
>
> Answers to your questions:
>
> 1. “What happens for optimizers that use adaptive learning rates, such as Adam?”
> That is an interesting question. In the paper, we consider SGD + momentum as a practical optimizer since it is a conventional choice for training convolutional neural networks. After your suggestion, we have conducted a supplemental experiment and trained a fully scale-invariant network using Adam using different values of learning rate and weight decay. The results are very similar to the momentum case: the periodic behavior occurs in a wide range of hyperparameter values but the periods are shorter than for the usual SGD with the same hyperparameter values. We will add these results to the supplementary materials.
> 2. “With what consistency does the loss achieve a lower minima after a phase of instability?”
> In our experiments, we have observed that during the warm-up stage the loss consistently achieves a lower minimum after a phase of instability. After the warm-up, when the weight norm converges to a stable range of values, the minimum loss value usually fluctuates around the same level: it can become a bit higher or a bit lower in the next training period.
> 3. Thank you for the suggestion, we will provide a more detailed explanation of this in the text. Making an SGD step in the direction of the loss gradient always increases the norm of scale-invariant parameters because the gradients are orthogonal to the radial direction (see property 2a). WD aims at decreasing the weight norm by definition.
>
> On minor comments:
>
> * “Using either epoch or SGD step (or making clearer how they relate to each other) consistently in the x-axis of figure 2+3”
> We use different values due to different experiment goals. In the majority of figures, we plot metric values computed after each epoch (e.g. Figure 2) while in Figure 3 we look at the effect in more detail and show metric values after each SGD step. In our experiments, one epoch consists of 391 SGD steps. We will clarify this in the text.
> * “Warm-up phases can't be seen too well from the plots”
> The distinctness of the warm-up period depends on the difference between the initial weight norm and the range of weight norms to which the training converges, therefore this period is not always present in the plots. We discuss this in more detail in Appendix E.1 (lines 670-676). In the presented figures, the warm-up period is most noticeable in the weight norm plots (e.g. red lines for lr = 0.03 in Figure 1).
> * “coarse grain choices of learning rate”
> We conducted all experiments with finer learning rate grain ($\\{10^{-k}, 3 \cdot 10^{-k}\\}_{k=0, 1, 2, 3}$) but we show only a subset of the plots that represents all types of behavior (we had to do so since it is difficult to distinguish between many periodic plots in one figure). However, while writing the paper we have analyzed the behavior of all the obtained plots, including not shown ones, and all of them match our conclusions. For example, for the experiment in Figure 2, the behavior of the line for lr = 0.003 is “between” lr = 0.01 and 0.001: it exhibits periodicity but with a very low frequency — in 1000 epochs it has only one destabilization around epoch 700.
> * “With which probability do the delta jumps that lead to instabilities actually occur? Do they happen with certainty if curve is in given interval, and does it always happen?”
> Our results are stated in a deterministic manner, i.e., we do not assume any probabilistic model, so, according to Proposition 1, the $\delta$-jump will *certainly* happen if the norm of the parameters is below $\frac{\eta \ell}{\sqrt{2 \delta}}$ and it will *never* happen if it is above $\frac{\eta L}{\sqrt{2 \delta}}$; once the parameters norm is between those values, $\delta$-jump may occur at any time depending on the local properties of loss landscape, the chosen optimizer, the behavior of gradients, etc.
> * “The description of the ensemble method in 5) was a little confusing”
> Thank you for the feedback, we will work more on this part of the text.

---

> > ### Comment · Reviewer_6wdK · 2021-08-31
> > **My concerns are addressed well**
> >
> > I really appreciate the authors’ detailed reply! They address my concerns. I have updated my score to reflect this. I hope I will  find all the mentioned modifications in the final version!

---

> > > ### Author Response · Authors · 2021-09-01
> > > **Thank you**
> > >
> > > Thank you! We are very glad that you were satisfied with our responses to your concerns, and we will definitely improve our text based on them.
> > >
> > > Best regards,
> > > Authors

---

### Author Response · Authors · 2021-08-10
**Authors general comment**

We are very grateful to all the reviewers for their constructive reviews! We appreciate that the reviewers found the discovered periodic behavior to be interesting and underlined the extensive experiments and clearly written text. We will incorporate your comments into the text that will definitely enhance the paper.

Since several reviewers pointed out that most of the experiments of the paper are conducted in a distilled scale-invariant setting, here we would like to note that the main motivation of such an experimental setup was to separate the effect of the interplay between normalized parameters and weight decay from the influence of other non scale-invariant weights, to provide the in-depth analysis of the reasons for the periodic behavior. Experiments from Section 6 and an additional experiment with Adam (see reply to question 1 of Reviewer 6wdK) show that the periodic behavior can be observed in practical scenarios as well if the network is trained long enough. We also think that studying such specific effects is important for understanding the general properties of neural networks training.

---

### Author Response · Authors · 2021-08-28
**Authors comment**

As the end of the discussion period draws near, we are looking forward to hearing your comments and are happy to address any remaining or new questions you may have.

Kind regards,
Authors

---

### Decision · Program_Chairs · 2021-09-27

**Decision:**

Accept (Poster)

**Comment:**

Three knowledgeable reviewers recommend this paper for acceptance and one more knowledgeable reviewer rates it as marginally below the acceptance threshold. After the discussion period, one reviewer concludes that although the theory part in the current form is not very strong, the empirical study in this paper is very nice and the periodic behaviour is very interesting. Another reviewer comments that s/he stands by her/his opinion that the discovery of the periodic behaviour is very interesting and warrants being shared with the NeurIPS community. Another reviewer indicates that although some of her/his main concerns still stand, s/he is convinced the extensive experiments are insightful and could be relevant in more practical cases and increased her/his score. In summary, after the rebuttal and discussion the general consensus among the reviewers is for accepting this paper. Hence I am recommending the paper for acceptance with the request that the authors take the reviewers’ comments and suggestions carefully into account for the preparation of the final manuscript.